# Host adaptive mutations in the 2009 H1N1 pandemic influenza A virus PA gene regulate translation efficiency of viral mRNAs via GRSF1

Michael Lutz [1], Jordana Schmierer[1] & Toru Takimoto [1✉]

Avian species are the major natural reservoir from which pandemic influenza A viruses can be introduced to humans. Avian influenza A virus genes, including the three viral polymerase genes, PA, PB1 and PB2, require host-adaptive mutations to allow for viral replication and transmission in humans. Previously, PA from the 2009 pH1N1 viral polymerase was found to harbor host-adaptive mutations leading to enhanced viral polymerase activity. By quantifying translation and mRNA transcription, we found that the 2009 pH1N1 PA, and the associated host-adaptive mutations, led to greater translation efficiency. This was due to enhanced cytosolic accumulation of viral mRNA, which was dependent on the host RNA binding protein GRSF1. Mutations to the GRSF1 binding site in viral mRNA, as well as GRSF1 knockdown, reduced cytosolic accumulation and translation efficiency of viral mRNAs. This study identifies a previously unrecognized mechanism by which host-adaptive mutations in PA regulate viral replication and host adaptation. Importantly, these results provide greater insight into the host adaptation process of IAVs and reveal the importance of GRSF1 in the lifecycle of IAV.

[1] Department of Microbiology and Immunology, University of Rochester Medical Center, Rochester, NY 14642, USA.
✉email: toru_takimoto@urmc.rochester.edu

nfluenza A viruses (IAVs) continue to be an important human pathogen capable of causing both seasonal epidemics and global pandemics. Avian species are the natural reservoir for IAV and avian IAVs represent a major source from which novel IAVs could be introduced to the human population, leading to a new pandemic[1,2]. However, avian IAVs must obtain host-adapting mutations in their viral RNA-dependent RNA polymerase (vRdRp) in order to efficiently replicate and transmit in humans[3–5]. The vRdRp is comprised of three components, PB1, PB2 and PA, bound to viral RNAs which are encapsulated by NP to form viral ribonucleoproteins. Various host-adaptive mutations have been identified in the vRdRp, mainly in PB2 and PA, suggesting that these two components play a key role[6–8].

Especially, the single mutation E627K in PB2 has been shown to significantly increase the activities of avian vRdRps in mammalian cells, leading to greater replication and pathogenicity of avian IAVs[9–12]. Recently, the mechanism by which PB2 E627K enhances the activity of avian vRdRps in mammalian cells has been elucidated and comes down to species-specific differences in ANP32 proteins, which facilitate vRdRp dimer formation required for genome replication[13–15]. Most mammalian ANP32A proteins lack a 33 amino acid insertion in the C-terminal low-complexity acidic region which is present in avian ANP32A proteins. Recent structural data has revealed that this 33 amino acid insertion, which is made up of both acidic and basic residues, in chicken ANP32A allows the C-terminal low-complexity acidic region to directly interact with the PB2 627 domain[16]. However, due to the lack of this 33 amino acid insertion in human ANP32A, the region of human ANP32A which contacts PB2 627 is almost entirely acidic, possibly giving an explanation for the strong selective pressure for a mutation from an acidic residue, glutamic acid (E), to a basic residue, lysine (K), at residue 627 in the PB2 of human adapted vRdRps[15].

The single PB2 E627K mutation was present in nearly all human IAVs since 1918 until the emergence of the 2009 pandemic H1N1 (pH1N1) virus, which lacked this mutation[17,18]. The vRdRp of the pH1N1 virus contained an avian-like PA and PB2 and was able to spread globally and cause disease[19]. Our lab discovered the PB2 mutation T271A had emerged with this pH1N1 virus and was able to activate avian vRdRp, possibly compensating somewhat for the lack of PB2 E627K[20]. Additionally, the mutation Q591K/R in PB2 has also been shown to play a role[21,22]. However, we and others found that the avian-like PA contributed the most to vRdRp activity of the pH1N1 virus in mammalian cells[6,20,23,24]. Previously, we identified PA mutations T85I, G186S and L336M in pH1N1, which enhanced avian IAV vRdRp activity in mammalian cells, as determined by reporter gene assay and mutant viruses[23]. In addition to these, more mutations in PA have been introduced into pH1N1 during seasonal circulation in the human population, including V100I[25]. However, the mechanism of how these PA mutations enhance vRdRp activity is not known. It is also unclear if these mutations directly affect the transcription and replication activities of the vRdRp or have alternate functions to enhance viral growth and protein production.

IAV is known to utilize several cellular RNA-binding proteins (RBPs) during its life cycle for various functions such as splicing, replication, and selective translation of viral mRNAs[26–28]. Among these RBPs is GRSF1, a member of the heterogeneous nuclear ribonucleoprotein (hnRNP) family[29]. GRSF1 has been shown to directly bind the sequence $^{10}$AGGGU$^{14}$ in the 5' UTR of IAV NP and NS mRNAs to enhance their translation efficiency[30–32]. GRSF1 has also been shown to bind similar sequences in other cellular mRNAs and regulate their translation, as well[33,34]. RBPs in the hnRNP family are known to regulate translation in a variety of ways, such as via mRNA trafficking, ribosome and translation factor recruitment, mRNA stability, and more[35]. However, it is not known if GRSF1-dependent selective expression of IAV proteins is a determinant for mammalian host adaptation.

Here, we followed up on our previous studies which identified several host-adaptive mutations in the PA from the 2009 pH1N1 virus. By examining the vRdRp activities of 2009 pH1N1 A/California/04/2009 (Cal), 2017 pH1N1 A/Michigan/272/2017 (Mich), and a prototypical avian strain A/chicken/Nanchang/3-120/01 (Nan), we found that mRNA transcripts produced by human adapted IAV vRdRps displayed greater translation efficiency. Specifically, the host-adaptive mutations T85I and G186S, and V100I, within the PA endonuclease domain were found to be involved in the regulation of cytosolic accumulation and translation of viral mRNA. The importance of mutations at residues 85 and 186 was confirmed in recombinant Cal pH1N1 viruses in which mutations to avian virus PA residues reduced viral protein synthesis, cytosolic accumulation of viral mRNAs, polysome association of viral mRNAs, and viral growth in mammalian cells. Furthermore, we found GRSF1 was required for the enhanced cytosolic accumulation and translation efficiency conferred by pH1N1 PA and the host-adaptive mutations T85I and G186S. Together, our results indicate that specific residues in PA affect the translation efficiency of viral mRNA via cytosolic accumulation which is dependent on the host RBP GRSF1.

## Results

**Origin of vRdRp components determines the translation efficiency of viral mRNA.** We previously found that the PA and PB2 subunits, but not PB1, from Cal independently and cooperatively enhance overall vRdRp activity of avian IAV as determined by reporter gene assays[20,23,36]. Here, we extended our reporter gene assays and measured both viral transcripts and translated products from the same cell lysates in order to understand the mechanism of enhanced vRdRp activity. Using a luciferase reporter gene, we quantitated both luciferase activity and mRNA production in cells transfected with wild type (WT) or mixed vRdRp cDNAs from the pH1N1 Cal strain and the avian strain Nan. Compared to Nan vRdRp, Cal vRdRp produced 24.2-fold more mRNAs in transfected cells (lane 1 vs 7, Fig. 1a). The Nan vRdRp containing Cal PA or Cal PB2 produced 26.6- or 13.2-fold more mRNAs than Nan vRdRp, confirming that both PA and PB2 genes of Cal enhance transcription activity of Nan polymerase (lanes 1–3, Fig. 1a). However, the enhancement of luciferase activity did not necessarily correlate with the quantity of mRNAs. Lysate of cells expressing Cal vRdRp showed 159.0-fold more luciferase activity compared to the Nan vRdRp (lane 1 vs 7, Fig. 1b). The Nan vRdRp containing Cal PA also showed a significant 257.6-fold increase in luciferase activity. However, Nan vRdRp containing Cal PB2 showed only 35.2-fold increase in luciferase activity (lanes 1–3, Fig. 1b). We calculated the ratios of luciferase activity per mRNA copy number, which we term here as translation efficiency. Comparison of the normalized ratios indicate that mRNA transcripts produced by Cal vRdRp, Nan vRdRp containing Cal PA alone or together with Cal PB2 have significantly greater translation efficiency than those produced by Nan vRdRp (lanes 1, 2, 4 and 7, Fig. 1c). Similarly, Cal vRdRp in which PA was replaced with that of Nan strongly reduced translational efficiency (lane 7 vs 6, Fig. 1c). Cal PB2 also appeared to affect translation efficiency (lanes 1 vs. 3 and 7 vs. 5, Fig. 1c), but this effect was not significant. We confirmed that firefly luciferase activity closely matched protein expression determined by immunoblotting, and there were no major differences in PA expression (Fig. 1d). These results indicate that the origin of PA strongly affects the translation efficiency of viral transcripts.

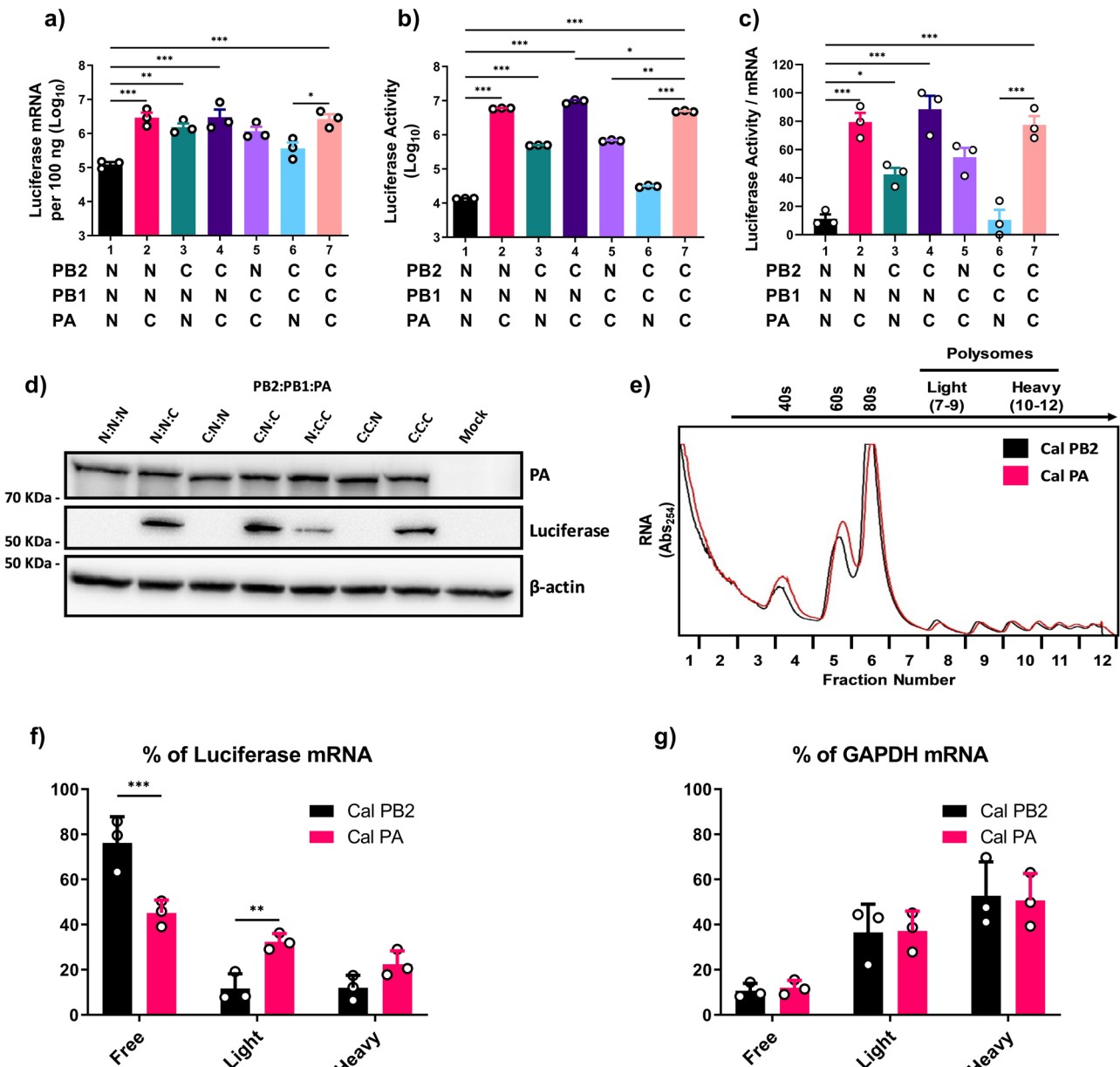

**Fig. 1 Origin of vRdRp components determines the translation efficiency of viral mRNA.** 293 T cells were transfected with the indicated viral polymerase subunits (N = Nan, C = Cal) along with pPoII-NP-luc. Cal NP was used in all conditions. **a** Luciferase mRNA from the pPoII-NP-luc construct was quantified by qRT-PCR. **b** Luciferase activity determined by the Dual-Luciferase Reporter Assay system (Promega). **c** Ratio of luciferase activity per mRNA, transformed to a normal distribution. **d** Representative image of immunoblot analysis of PA, luciferase and β-actin in cell lysates using specific antibodies. **e**–**g** 293 T cells were transfected with either Nan NP, PB1, PB2 and Cal PA or Nan NP, PB1, PA, and Cal PB2 along with pPoII NP-luc. **e** Representative polysome traces for the two conditions are overlaid with one another. **f**, **g** mRNA from pPoII-NP-luc and cellular GAPDH was quantified from each fraction by qRT-PCR and grouped into polysome free (fractions 1–6), light (fractions 7–9) or heavy (fractions 10–12). All error bars show means plus/minus the standard deviations (n = 3 biological replicates). One-way ANOVA followed by Tukey's multiple comparison test (*$P < 0.05$, **$P < 0.01$, ***$P < 0.001$).

**PA impacts the ability of viral mRNAs to form polysomes.** Transcripts which are actively translated are associated with multiple ribosomes and form polysomes, allowing for multiple proteins to be rapidly translated from a single mRNA[37–40]. Therefore, we examined the association of viral mRNAs with ribosomes by polysome fractionation assay. We analyzed the transcripts produced by Nan vRdRp containing either PA or PB2 from Cal. These polymerase complexes produced similar levels (<2-fold difference) of transcripts, but showed a large difference in translation efficiency (lane 2 vs 3, Fig. 1a and c). Transfected cell lysates were ultra-centrifuged through sucrose gradients, fractionated, and mRNAs in each fraction were quantitated by qRT-PCR (Fig. 1e). When the quantity of polysome associated mRNAs were compared, we found that more mRNAs produced by Nan vRdRp with Cal PA were recovered from polysome fractions (fractions 7–12) compared to those produced by Nan vRdRp with Cal PB2 (54.9% and 23.8%, respectively, Fig. 1f). Similarly, 13.4% of the transcripts produced by vRdRp with Cal PA were recovered from the fraction containing 80 S ribosome (fraction 6), while only 5.4% transcripts from vRdRp with Cal PB2 were recovered from this fraction. These results are consistent with the data showing efficient translation of transcripts produced by Nan vRdRp containing Cal PA (Fig. 1c).

The observed differences in translation efficiency could be due to the global effect of vRdRp proteins on host translational

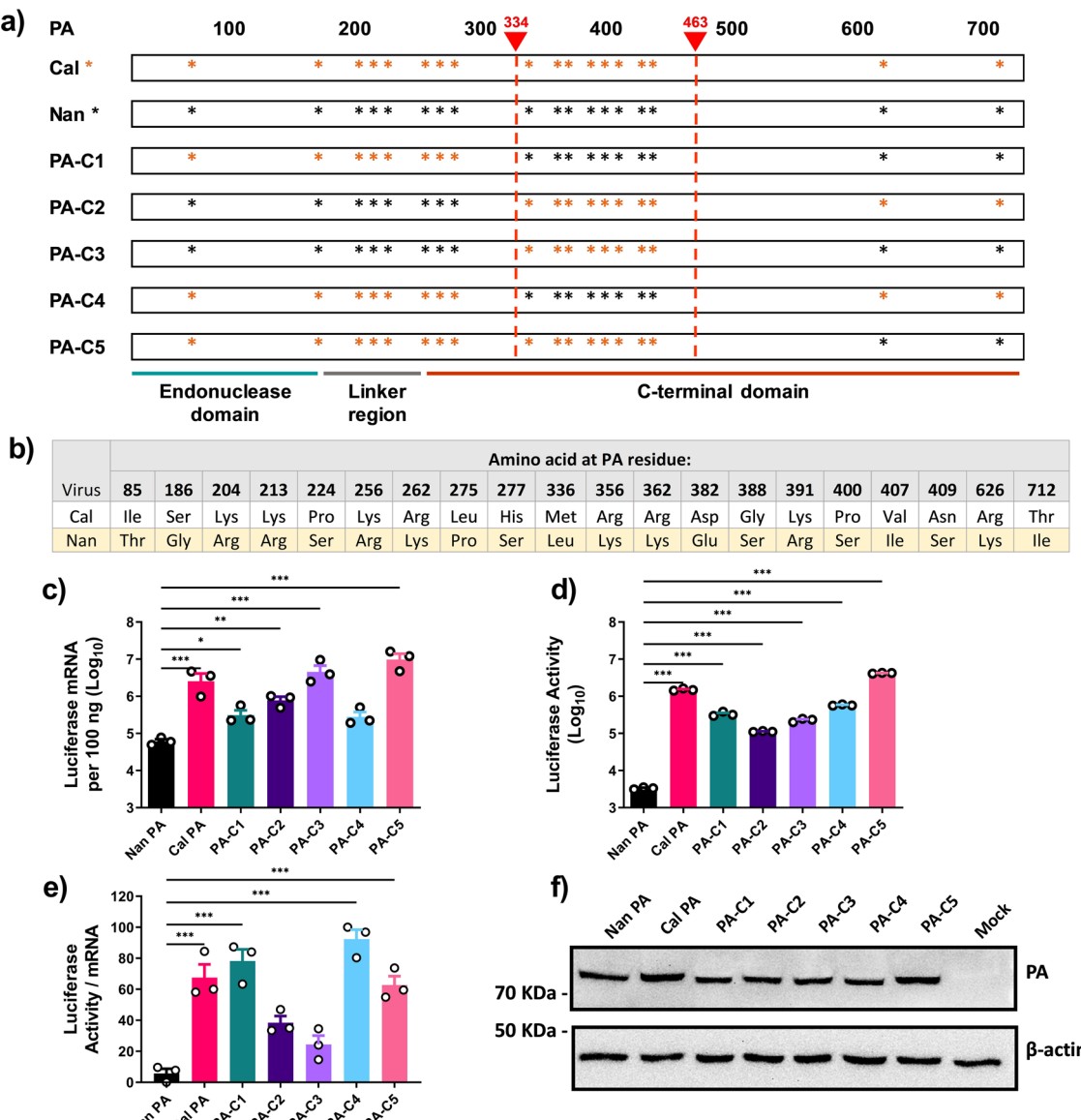

**Fig. 2 The N-terminal region of Cal PA enhances the translation efficiency of viral mRNA. a** Diagram showing the chimeric PA constructs. The asterisks indicate differences in amino acids. **b** Amino acid differences between Cal and Nan PA. **c–f** 293 T cells were transfected with expression plasmids for Nan NP, PB1, PB2 and the indicated PA subunits along with pPolI-NP-luc. **c** Luciferase mRNA was quantified by qRT-PCR. **d** Luciferase activity determined by the Dual-Luciferase Reporter Assay system. **e** Ratio of luciferase activity per mRNA, transformed to a normal distribution. **f** Representative image of immunoblot analysis of PA and β-actin in cell lysates using specific antibodies. All error bars show means plus/minus the standard deviations ($n = 3$ biological replicates). One-way ANOVA followed by Tukey's multiple comparison test (*$P < 0.05$, **$P < 0.01$, ***$P < 0.001$).

machinery. To determine this, we quantitated host GAPDH mRNAs in each fraction of the same samples. We found that GAPDH mRNAs efficiently recruit ribosomes and form polysomes, and there was no difference between the cells transfected with the different vRdRp complexes (Fig. 1g).

**The PA N-terminal region affects the translation efficiency of viral mRNA.** To determine which region(s) within PA are involved in regulating translation efficiency, we characterized the activities of chimeric PAs containing sequences from Cal and Nan PA (Fig. 2a, b). We expressed these chimeric PA genes together with Nan NP, PB1, PB2 and reporter gene and quantitated luciferase activity and transcripts. When comparing mRNA production, the chimeras containing amino acids 334–463 from Cal PA showed higher transcription activity than those

containing the same region from Nan PA (PA-C2; 12.9-fold, PA-C3; 75.9-fold, and PA-C5; 162.2-fold compared to Nan PA, Fig. 2c). These differences were not due to differential expression of the PA chimeras since similar levels of the PAs were detected in transfected cells by immunoblotting (Fig. 2f). For luciferase activity, the chimeras containing the PA N-terminal 333 residues from Cal produced the most activity compared to Nan (PA-C1; 100.0-fold, PA-C4; 173.8-fold and PA-C5; 1,258.9-fold, Fig. 2d). When compared to the mRNA transcript production, the luciferase activity again did not necessarily correlate. The translation efficiency was significantly higher for those produced by vRdRp containing the PA N-terminal 333 residues from Cal compared to Nan (PA-C1, PA-C4, and PA-C5, Fig. 2e). By comparison, the PA chimeras containing the same N-terminal region from Nan PA did not have significantly greater efficiency than that of WT Nan PA (PA-C2 and PA-C3, Fig. 2e). These data indicate that different

regions of PA affect the transcription activity and translation efficiency of the transcripts, and that the N-terminal region including the endonuclease domain strongly affects the translation efficiency of viral mRNAs.

**Residues 85, 100, and 186 are involved in the regulation of translation efficiency.** Results of the study using chimeric PA indicate that the residues within the N-terminal region are responsible for enhanced translation (Fig. 2). Previously, we identified two mutations T85I and G186S in the N-terminal region of Cal PA, which are involved in increasing vRdRp activity[23]. These residues are located within the endonuclease domain of PA, which is associated with cap snatching and initiation of transcription[41,42]. Additionally, since 2009, the pH1N1 viruses continued to circulate in humans and gained additional mutations, some of which are within the polymerase genes[43–45]. We characterized the polymerase genes of a pH1N1 strain isolated in 2017, A/Michigan/272/2017 (Mich), and found that Mich PA was also able to further enhance the translation efficiency of the Cal vRdRp (Supplementary Fig. 1). Mich contains an additional 6 mutations in PA (residues 100, 224, 321, 330, 362, 438), one of which, V100I, is within the PA endonuclease domain and subsequently was included in our study alongside residues 85 and 186 (Fig. 3a).

We next determined the effect of the three mutations within the endonuclease domain, T85I, G186S, and V100I. Nan PA containing each mutation individually or in combinations were tested by reporter gene assay together with the rest of the components from Nan. All mutant PAs were expressed well as determined by immunoblot analysis (Fig. 3h). Individual or combined mutations in Nan PA showed no or slight increase of transcription activity (0.9–2.5 fold) compared to WT (Fig. 3b). However, we did detect greater variations in luciferase activity ranging from 0.98–7.4 fold (Fig. 3c). Among the single residue mutations, V100I significantly increased translation efficiency (Fig. 3d). Individual mutations of T85I and G186S did not significantly affect translation efficiency; however, the combination of these two mutations did (Fig. 3d). These results suggest that all three residues contribute to regulating the translation efficiency of mRNAs, and that residues 85 I and 186 S likely act cooperatively.

We then tested whether mutations to Nan residues at 85 and 186 in Cal PA reduce translation efficiency and if V100I could further increase translation efficiency. Again, all of the mutant PAs were found to be expressed well (Fig. 3i). The individual reversion mutations, I85T and S186G did not significantly reduce transcriptional activity, but the combination of both did by 5.5-fold (Fig. 3e). A similar, albeit more pronounced, trend was observed for luciferase activity where the combination of I85T and S186G caused a 102.3-fold decrease in activity (Fig. 3f). This combined mutant led to a significant decrease in translation efficiency (Fig. 3g). In contrast, the mutation V100I in Cal PA significantly enhanced luciferase activity by 2.7-fold with no significant effect on mRNA transcription. This mutation did lead to a slight, albeit non-significant increase in translation efficiency (Fig. 3f). The data suggest these mutations, most specifically T85I and G186S, obtained by pH1N1 PA contribute to the vRdRp being able to produce viral mRNAs which are efficiently translated within infected cells.

**Viral mRNAs transcribed by vRdRp containing Nan PA do not accumulate in the cytosol.** In order to determine the mechanism by which PA residues in the endonuclease domain enhanced translation efficiency, we considered multiple possibilities. First, the key residues identified are located within the endonuclease domain, which is shared with PA-X. PA-X is a host shutoff protein produced by ribosomal frameshifting and shares the N-terminal 191 residues with PA[46–48]. PA-X suppresses host gene expression by targeting host RNA Pol II transcripts for degradation via its endonuclease domain[49]. Therefore, it is possible that mutations in the shared domain between PA and PA-X enhance PA-X activity to degrade more host mRNAs and allow viral transcripts better access to ribosomes and translational machinery, resulting in enhanced translation of viral transcripts. We tested the shutoff activity of Nan or Cal PA-X containing the aforementioned mutations and in fact, the opposite was observed (Fig. 4). The presence of the avian-like residues I85T and S186G in Cal PA-X significantly increased shutoff activity while decreasing the translation efficiency (Figs. 4a and 3g). In agreement with this we found that the presence of both T85I and G186S in Nan PA-X significantly reduced shutoff activity while contributing to increased translation efficiency (Figs. 4b and 3d). To further eliminate the possibility that the activity of PA-X plays a role in translation efficiency, we created Nan PA cDNA containing silent mutations at the frameshift site to reduce the expression of PA-X as described previously[48]. In reporter gene assays using Nan PA frameshift mutants, we still detected differences in translation efficiency, suggesting that PA-X activity is not responsible for the enhanced translation of viral mRNAs (Fig. 4c–e).

Next, we sought to determine if the cytosolic levels of viral mRNA was affected by the origin of PA. The IAV vRdRp carries out transcription within the host cell nucleus which necessitates the export of viral mRNAs to the cytosol for translation[50,51]. Cells transfected with WT or mixed vRdRps from Cal or Nan were fractionated into nuclear and cytosolic fractions, and vRdRp transcribed mRNAs were quantified by qRT-PCR. Strikingly, we detected major differences in the cytosolic levels of viral mRNA that depended on the origin of the vRdRp (Fig. 5a). Only 2.35% of mRNAs transcribed by Nan vRdRp were detected in the cytosol in contrast to 24.49% of mRNAs transcribed by Cal vRdRp. Importantly, 20.37% of mRNAs produced by Nan vRdRp containing Cal PA were detected in the cytosol and, similarly, the presence of Nan PA in Cal vRdRp reduced the percentage of cytosolic mRNA to 7.12% (Fig. 5a). These results clearly indicate that PA determined the efficiency of mRNA accumulation in the cytosol. These effects were specific to vRdRp transcribed mRNA as the percentage of cytosolic GAPDH mRNA and nuclear U6 snRNA was similar for all conditions (Fig. 5b). Immunoblotting for lamin (nuclear marker) and tubulin (cytosol marker) confirmed the specificity of the fractionation (Fig. 5c). Furthermore, host-adaptive mutations, V100I, and T85I and G186S, in Nan PA enhanced the accumulation of Nan vRdRp transcribed mRNAs in the cytosol (Fig. 5d). This effect was found to be greater in the context of the Cal vRdRp with Nan PA, suggesting that other components of the vRdRp also play a role and may act synergistically with these host-adaptive mutations (Fig. 5e). These results suggest that host-adaptive mutations in PA have a strong influence on the cytosolic accumulation of viral mRNAs, possibly via nuclear export.

**GRSF1 binding regulates cytosolic accumulation and translation efficiency of viral mRNA.** Previous studies have shown that the host RBP GRSF1 attaches to IAV mRNAs and enhances the translation of viral transcripts as determined in vitro using IAV infected cell lysates[30]. Specifically, it was shown that GRSF1 binds to a conserved [10]AGGGU[14] sequence in the 5' UTR of NP and NS IAV mRNAs[31]. Notably, this GRSF1 binding sequence is not conserved in other IAV mRNAs, such as PB1, where instead the sequence reads [10]AGGCA[14](Supplementary Fig. 2 and 3). Our

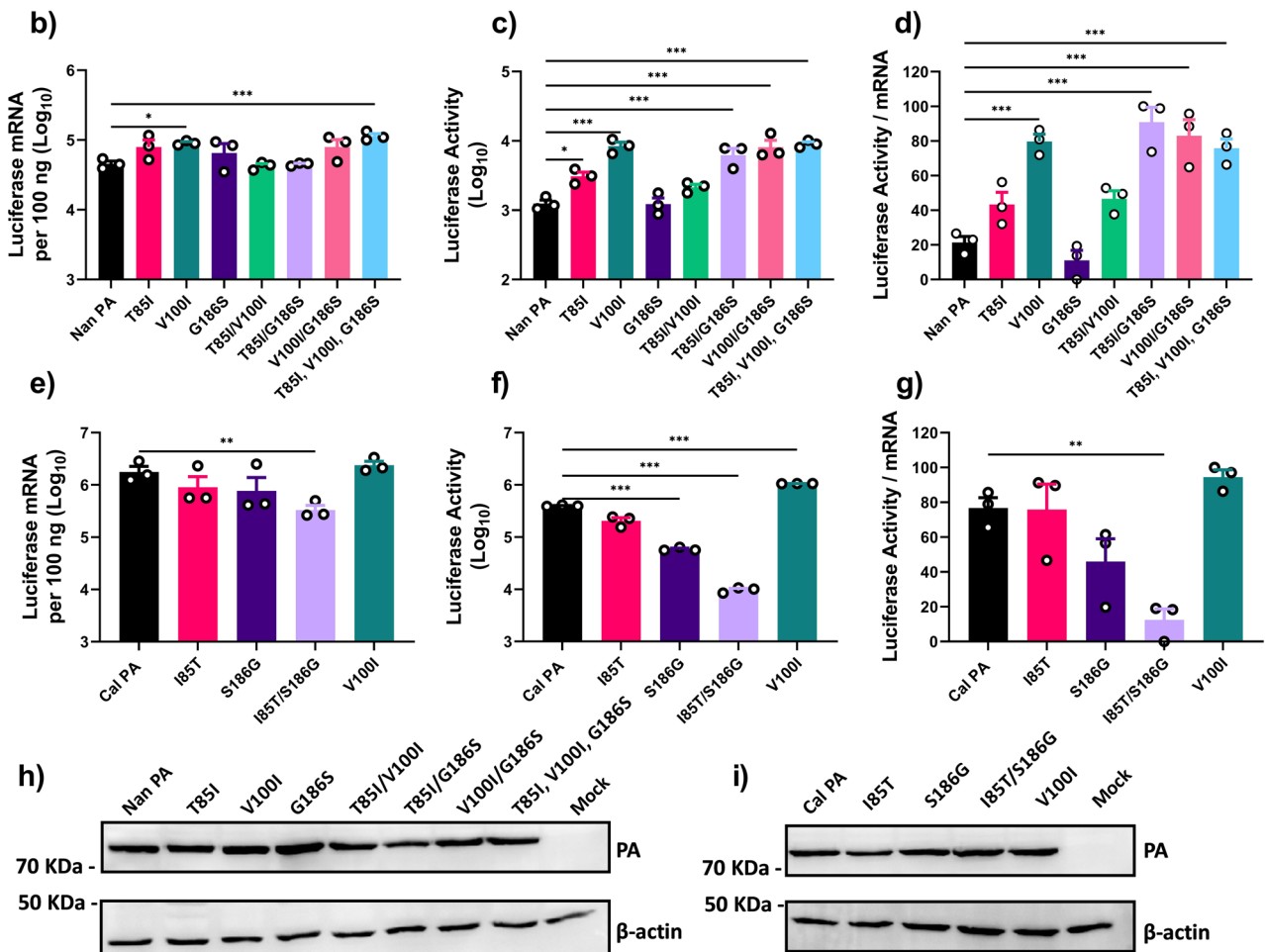

| Identity of Residue 100 in pH1N1 PA from 2009 - 2020 | | | | | | | | | | | | |
|---|---|---|---|---|---|---|---|---|---|---|---|---|
|  | 2009 | 2010 | 2011 | 2012 | 2013 | 2014 | 2015 | 2016 | 2017 | 2018 | 2019 | 2020 |
| % Valine | 99.22 | 99.74 | 99.58 | 93.51 | 65.18 | 2.59 | 16.44 | 1.86 | 0.40 | 0.00 | 0.08 | 0.00 |
| % Isoleucine | 0.59 | 0.26 | 0.21 | 5.95 | 34.19 | 96.98 | 83.56 | 97.71 | 98.80 | 98.65 | 98.85 | 96.93 |

**Fig. 3 PA mutations T85I, V100I and G186S increase the translation efficiency. a** Percentages of pH1N1 viruses containing valine or isoleucine at residue 100 in PA from 2009 to 2020. Sequences of pH1N1 viruses were downloaded from fludb.org for comparison. **b–d** 293 T cells were transfected with expression plasmids for Nan NP, PB1, PB2 and Nan PA with the indicated mutations along with pPolI-NP-luc. **e–g** 293 T cells were transfected with expression plasmids for Cal NP, PB1, PB2 and Cal PA with the indicated mutations along with pPolI-NP-luc. **b**, **e** Luciferase mRNA was quantified by qRT-PCR. **c**, **f** Luciferase activity determined by the Dual-Luciferase Reporter Assay system. **d**, **g** Ratio of luciferase activity per mRNA, transformed to a normal distribution. **h**, **i** Representative images of immunoblot analysis of PA and β-actin in cell lysates using specific antibodies. Error bars show means plus/minus the standard deviations ($n = 3$ biological replicates). One-way ANOVA followed by Tukey's multiple comparison test (*$P < 0.05$, **$P < 0.01$, ***$P < 0.001$).

pPolI-NP-luc reporter contains the 5′ and 3′ UTRs from the NP gene of A/WSN/1933(H1N1), which includes the conserved GRSF1 binding sequence. To determine if GRSF1 binds mRNA produced by the vRdRp from pPolI-NP-luc, and if we could abolish this binding by mutating the $^{10}$AGGGU$^{14}$sequence to $^{10}$AGGCA$^{14}$, we performed RNA immunoprecipitation analysis (Fig. 6a). Anti-GRSF1 or IgG rabbit control antibodies were used for immunoprecipitation on lysates from transfected cells (Supplementary Fig. 4a) and recovered luciferase reporter mRNAs were quantitated by qRT-PCR.

We found that mRNA transcribed by Cal vRdRp was bound by GRSF1 at a higher rate compared to mRNA transcribed by Nan vRdRp, and importantly mutation of the GRSF1 binding

sequence to $^{10}$AGGCA$^{14}$ (CA Mutant) significantly decreased the mRNA which was bound by GRSF1 (Fig. 6b). Next, we performed the same experiment utilizing the Cal vRdRp with Nan PA or Nan PA with the indicated mutations to compare the translation efficiencies of the original and CA mutant luciferase mRNAs (Fig. 6c–e). Transcription of the CA mutant luciferase mRNA was not found to be different from the original template (Fig. 6c). However, luciferase production and translation efficiency of CA mutant luciferase mRNA was significantly lower than that observed with the original luciferase mRNAs (Fig. 6d, e). To determine if the CA mutation at the GRSF1 binding site affected the cytosolic levels of vRdRp transcribed mRNA, we performed nuclear/cytosol fractionation and qRT-PCR as

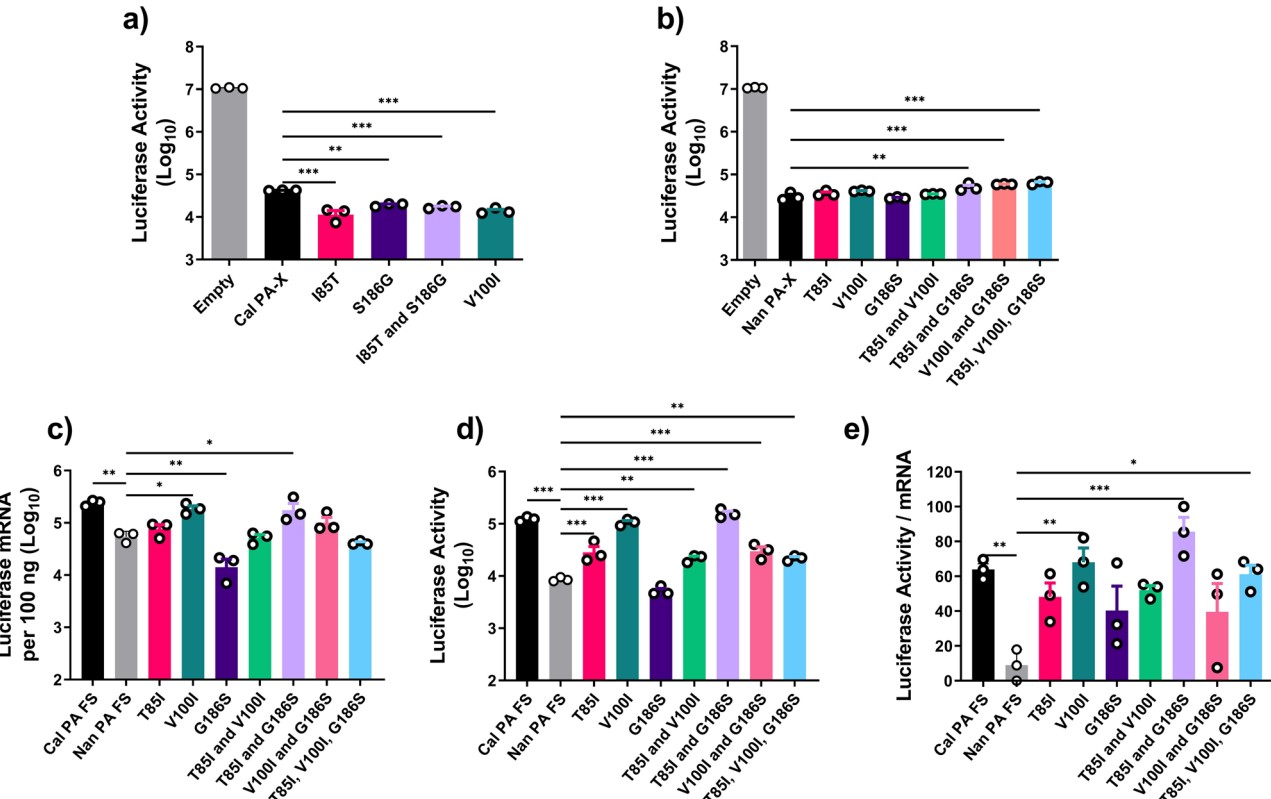

**Fig. 4 Shutoff activity of PA-X does not affect the translation efficiency of viral mRNA. a**, **b** Shutoff activity of Cal and Nan mutant PA-Xs. 293 T cells were transfected with expression plasmids for Cal PA-X **a** or Nan PA-X **b** with the indicated mutations along with pCAGGS firefly luciferase. Luciferase activity was determined by the Dual-Luciferase Reporter Assay system. **c–e** Translation efficiency of mRNAs produced by vRdRps containing PA frameshift mutant. 293 T cells were transfected with expression plasmids for Nan NP, PB1, PB2 and Cal PA FS (frameshift) or Nan PA FS with the indicated mutations along with pPolI-NP-luc. **c** Luciferase mRNA was quantified by qRT-PCR. **d** Luciferase activity was determined by the Dual-Luciferase Reporter Assay system. **e** Ratio of luciferase activity per mRNA, transformed to a normal distribution. Error bars show means plus/minus the standard deviations ($n = 3$ biological replicates). One-way ANOVA followed by Tukey's multiple comparison test (*$P < 0.05$, **$P < 0.01$, ***$P < 0.001$).

described above. Similar to translation efficiency, we found that the percentage of cytosolic vRdRp transcribed mRNA was decreased in all conditions which used the CA mutant reporter gene compared to the original (Fig. 6f). The same results were obtained when mixed vRdRp containing Cal or Nan components were used (Supplementary Fig. 4b–e). It should be noted however, even with the CA mutant GRSF1 binding site, mutations V100I, and T85I and G186S in Nan PA slightly increased the translation efficiency and cytosolic levels of luciferase mRNA in complex with Cal PB1 and PB2 components, suggesting that these mutations may act synergistically with other components of the Cal vRdRp.

**Knockdown of GRSF1 reduces cytosolic accumulation and translation efficiency of viral mRNA.** To further confirm the role of GRSF1 in cytosolic accumulation and translation efficiency of IAV mRNAs, we established GRSF1 knockdown (GRSF1 KD) cell lines that stably express shRNA for all isoforms of GRSF1. We confirmed reduced protein and mRNA expression of GRSF1 in two KD cell lines (clones 1–2 and 1–16) (Fig. 7a, b). We also confirmed that the nuclear/cytosolic localization of mRNAs and proteins of representative host genes was not affected by knockdown of GRSF1 in these cell lines (Fig. 7c, d).

We then preformed reporter gene assays in these two GRSF1 KD cell lines with the original pPolI-NP-luc reporter to determine the effect of GRSF1 depletion on translation efficiency and cytosolic levels of vRdRp transcribed mRNAs. Reporter gene assays performed in these GRSF1 KD cell lines utilizing the Cal

vRdRp with Nan PA or Nan PA with the indicated host-adaptive mutations showed that transcription was only slightly reduced in one of the GRSF1 KD cell lines (Fig. 7e, Clone 1–2). Importantly, luciferase activity and translation efficiency were significantly reduced in both GRSF1 KD cell lines (Fig. 7f, g). In addition, the quantity of vRdRp transcribed mRNA in the cytosol was significantly reduced in the GRSF1 KD cells compared to that in WT cells (Fig. 7h). Overall, these data further support the idea these host-adaptive mutations in PA are heavily reliant on host GRSF1 for accumulation in the cytosol and translation of vRdRp transcribed mRNAs.

**PA mutations and GRSF1 regulate viral growth, cytosolic mRNA levels, and protein expression.** To verify the effect of PA mutations on cytosolic accumulation and translation efficiency of viral mRNAs in the context of viral infection, we rescued pH1N1 Cal viruses which contained the avian-like mutations I85T and S186G in combination and a virus containing the mutation V100I. We were unable to rescue recombinant Cal virus with the entire Nan PA gene despite multiple attempts possibly due to significant reduction in polymerase activity (Fig. 1). Using the rescued viruses, we first conducted a multi-step growth curve in human lung epithelial Calu-3 cells. Compared to WT Cal, the PA V100I mutant grew faster and to slightly higher titers, most notably at 6 h post infection (hpi) (Fig. 8a). In contrast, mutations to avian virus residues in PA were severely detrimental to virus growth as demonstrated by the I85T and S186G virus which grew much slower and to lower titers than WT Cal (Fig. 8a).

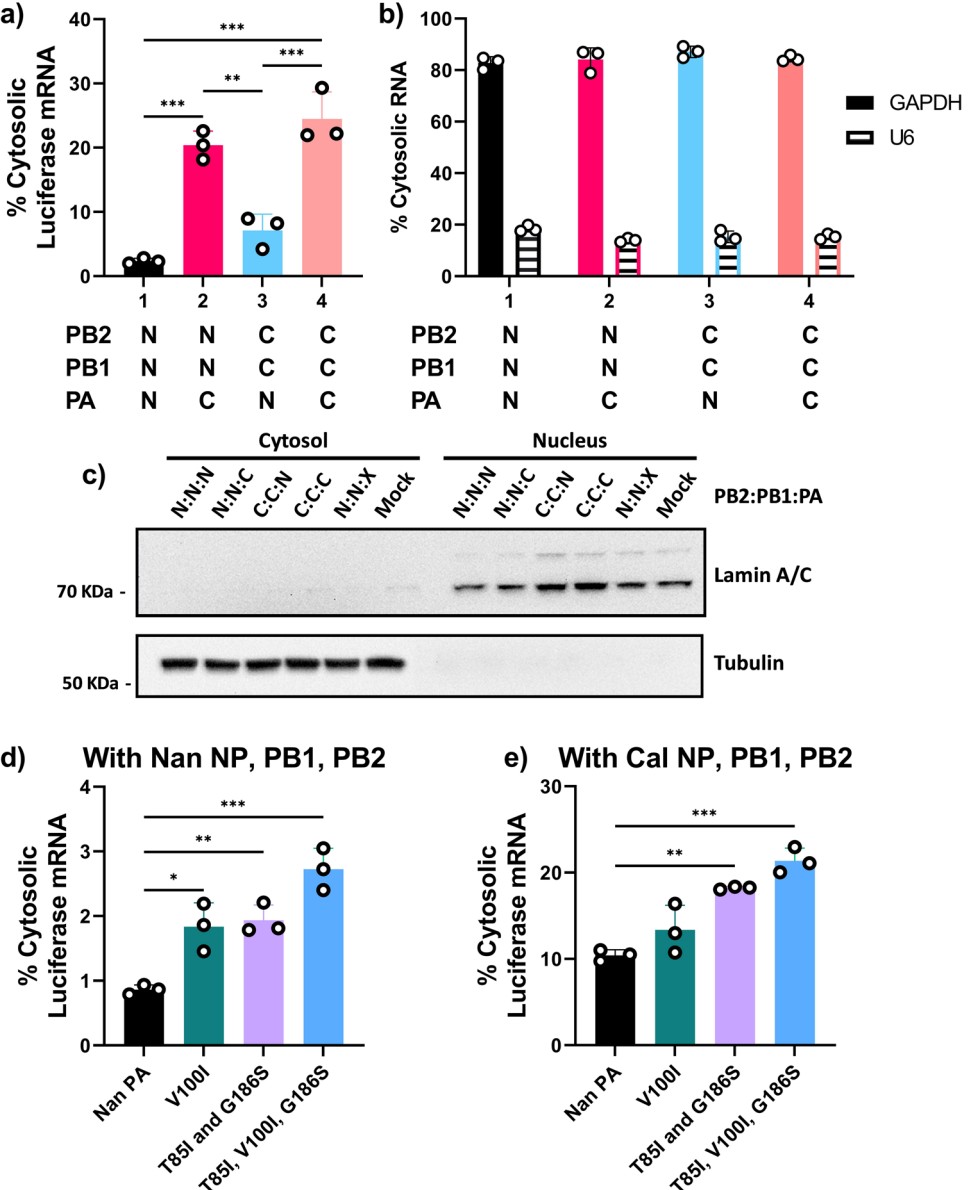

**Fig. 5 PA mutations affect the cytosolic levels of viral mRNA. a–c** 293 T cells were transfected with the indicated vRdRp subunits (N = Nan, C = Cal) along with Cal NP and pPolI-NP-luc. Cells were fractionated into nuclear and cytosolic fractions and RNAs were quantitated by qRT-PCR. **a** Percentages of cytosolic luciferase mRNA. **b** Percentages of cytosolic GAPDH mRNA and U6 snRNA. **c** Representative image of immunoblot analysis for tubulin (cytosolic marker) and lamin (nuclear marker) detected by specific antibodies. X indicates no PA. **d** 293 T cells were transfected with Nan NP, PB1, PB2, pPolI-NP-luc and Nan PA with the indicated mutations. Cells were fractionated into nuclear and cytosolic fractions and RNAs were quantitated by qRT-PCR. **e** 293 T cells were transfected with Cal NP, PB1, PB2, pPolI-NP-luc and Nan PA with the indicated mutations. Cells were fractionated into nuclear and cytosolic fractions and RNAs were quantitated by qRT-PCR. Error bars show the means plus/minus standard deviations (n = 3 biological replicates). One-way ANOVA followed by Tukey's multiple comparison test (*P < 0.05, **P < 0.01, ***P < 0.001).

To estimate translation efficiency, we determined the association of viral mRNAs with ribosomes in infected cells by polysome fractionation assay. Consistent with the viral growth data, we found greater amounts of NP mRNA associated with heavy polysomes for the WT Cal and V100I viruses compared to the I85T and S186G virus (Fig. 8b). Next, protein synthesis in infected cells was determined by metabolic labeling (Fig. 8c). The results showed that the V100I mutation led to greater synthesis of viral proteins, most notably HA, NP, M1, and NS1, compared to WT Cal. The I85T and S186G virus consistently displayed reductions in viral protein synthesis (Fig. 8c).

Lastly, we characterized the effect of GRSF1 KD on viral mRNA during infection. WT and GRSF1 KD cells were infected

and total mRNA for NP and PB1 was quantified. Similar to what was seen in reporter gene assays, total NP mRNA transcription was only slightly decreased in one GRSF1 KD cell line (Fig. 8d). Importantly, levels of cytosolic NP mRNAs were significantly reduced in both GRSF1 KD cell lines for the WT and V100I viruses (Fig. 8f). NP mRNAs produced by the virus with avian-like mutations I85T and S186G were not efficiently translocated to the cytosol, even in the presence of GRSF1. These data agree with our reporter gene assays in the GRSF1 KD cells and suggest that, for NP mRNA, residues 85 and 186 are somewhat reliant on GRSF1 for cytosolic accumulation, possibly via nuclear export. In contrast, despite seeing significant reductions in PB1 mRNA in the GRSF1 KD cells (Fig. 8e), the cytosolic levels of PB1 mRNA

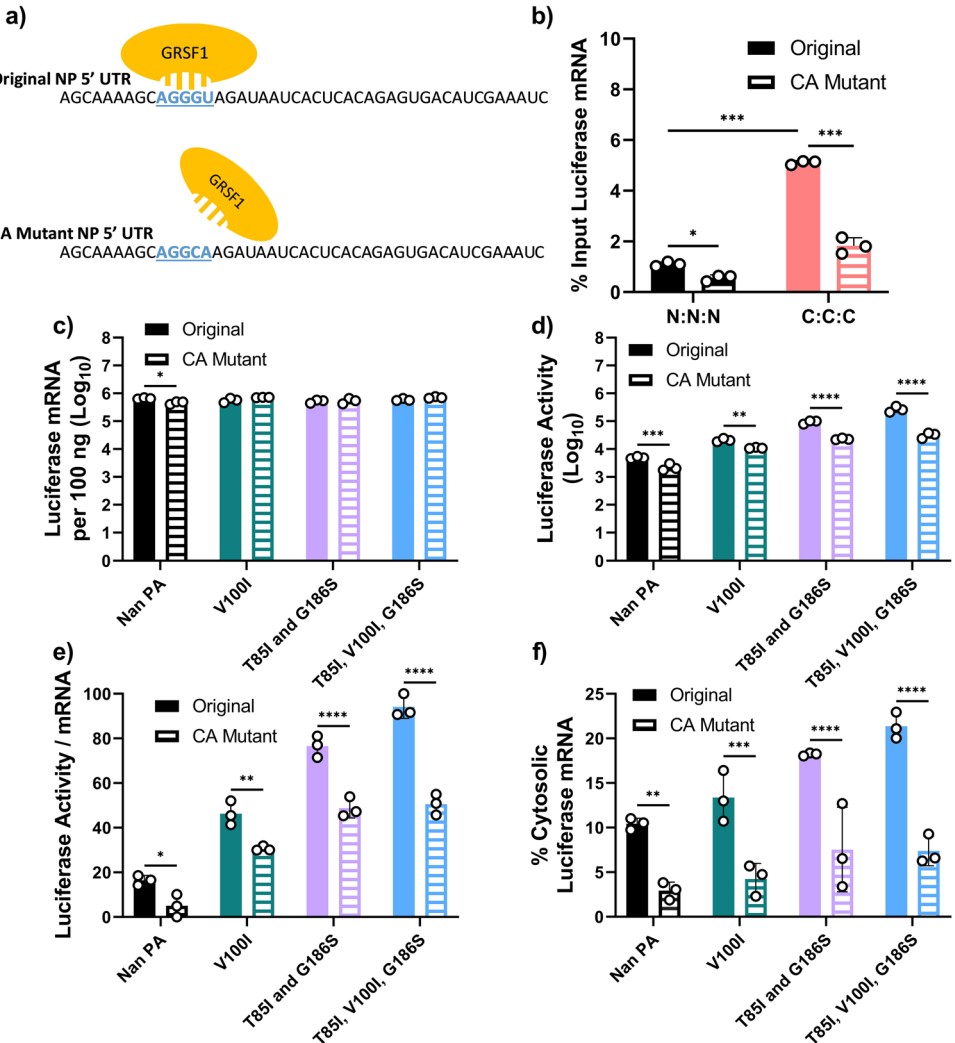

**Fig. 6 GRSF1 binding regulates cytosolic accumulation and translation efficiency. a** A model diagram showing the interaction of GRSF1 with the [10]AGGGU[14] binding site in the 5' UTR of NP luciferase mRNA. **b** 293 T cells were transfected with Cal NP and Nan or Cal vRdRp components along with the original pPolI-NP-luc or pPolI-NP-luc CA mutant. Anti-GRSF1 rabbit antibody was used for immunoprecipitation (IP) from whole cell lysates and RNA was extracted from the eluted material. Luciferase mRNAs before and after IP were quantitated by qRT-PCR and percentages of mRNA recovered after IP are shown. **c**, **d** 293 T cells were transfected with Cal NP, PB1, PB2, and either Nan PA or Nan PA mutants together with the original pPolI-NP-luc or the pPolI-NP-luc CA mutant **c** Luciferase mRNA was quantitated by qRT-PCR. **d** Luciferase activity was determined by the Dual-Luciferase Assay Reporter system. **e** Ratio of luciferase activity per mRNA, transformed to a normal distribution. **f** Percentages of cytosolic luciferase mRNA for both templates. Error bars show the means plus/minus standard deviations ($n = 3$ biological replicates). Two-Way ANOVA followed by Tukey's multiple comparison test (*$P < 0.05$, **$P < 0.01$, ***$P < 0.001$).

were largely unaffected by GRSF1 depletion, indicating that GRSF1 is not involved in the cytosolic accumulation of PB1 mRNA (Fig. 8g). Overall, these data demonstrate a link between residues 85 and 186 and host factor GRSF1 in facilitating the efficient cytosolic accumulation and translation of viral mRNAs, which has important consequences for viral fitness.

## Discussion

It has been well recognized that vRdRp mutations are necessary for avian IAVs to cross the species barrier and productively infect humans. Many host-adaptive mutations of the vRdRp have been identified, mainly in PA and PB2[3]. However, the underlying mechanisms of how these mutations activate vRdRp activity in mammalian hosts have not been fully elucidated, with the recent exception of PB2 E627K. On the other hand, there exists little mechanistic explanations for how host adaptive mutations in PA regulate vRdRp activity. Mostly, these mutations in PA are

located within either the endonuclease domain, which is required for cap-snatching to initiate viral mRNA transcription, or the C-terminal domain, which is involved in the interaction with host RNA polymerase II or vRdRp dimerization[52–54]. Here, we have uncovered a mechanism by which host-adaptive mutations in the endonuclease domain of PA regulate the cytosolic accumulation and translation of viral mRNAs via the host factor GRSF1. By quantifying vRdRp produced transcripts and their subsequently expressed proteins, we have revealed that host-adaptive mutations in PA accelerate viral protein production by enhancing the translation efficiency of transcripts rather than simply increasing their quantity. This appears to affect only a subset specific viral mRNAs which can be bound by the host factor GRSF1. This is a previously unrecognized mechanism of host adaptation by the IAV vRdRp which is independent of transcriptase or replicase activity.

Specifically, what we have shown is that viral mRNAs produced by vRdRps containing host-adaptive mutations T85I and G186S

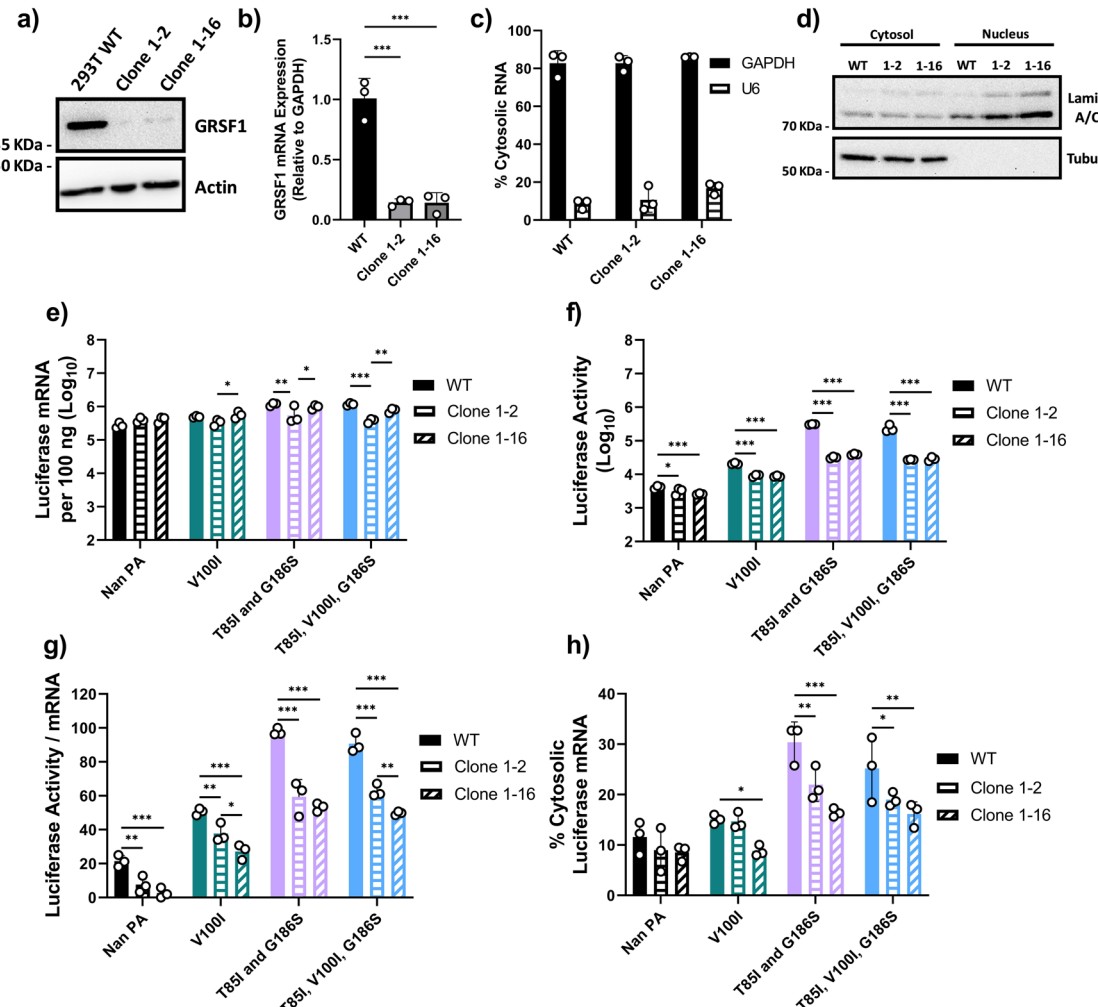

**Fig. 7 GRSF1 knockdown reduces cytosolic accumulation and translation efficiency. a** Lysates from WT 293 T or 293 T cells stably expressing shRNA for GRSF1 (clone 1–2 and clone 1–16) were used for immunoblotting to determine GRSF1 and β-actin expression. **b** GRSF1 mRNA expression from total cell lysates was determined by qRT-PCR using the 2^ΔΔ Ct method. **c, d** WT 293 T or shRNA GRSF1 clone 1–2 or 1–16 cells were fractionated into nuclear and cytosolic fractions. **c** Percentages of cytosolic GAPDH mRNA and U6 snRNA are shown. **d** Representative image of immunoblotting analysis for tubulin (cytosolic marker) and lamin (nuclear marker) detected by specific antibodies. **e–h** WT 293 T or shRNA GRSF1 clone 1–2 or 1–16 cells were transfected with Cal NP, PB1, PB2, pPolI-NP-luc and Nan PA with the indicated mutations. **e** Luciferase mRNA was quantified by qRT-PCR. **f** Luciferase activity was determined by the Dual-Luciferase Reporter Assay system. **g** Ratio of luciferase activity per mRNA, transformed to a normal distribution. **h** Cells were fractionated into nuclear and cytosolic fractions and mRNA was quantified by qRT-PCR. Percentages of cytosolic luciferase mRNA are shown. Error bars show the means plus/minus standard deviations ($n = 3$ biological replicates). One-way or Two-way ANOVA followed by Tukey's multiple comparison test $*P < 0.05$, $**P < 0.01$, $***P < 0.001$.

in PA were able to efficiently accumulate in the cytosol and be translated into protein. This effect was dependent on the host RBP GRSF1. Knockdown of GRSF1 significantly reduced translation efficiency and the cytosolic accumulation of viral mRNA. It is of note that the major isoform of GRSF1 is predominantly localized to mitochondria where it has been shown to regulate RNA processing and degradation[55,56]. Therefore, knockdown of GRSF1 expression, as preformed here, could lead to release of RNAs from the mitochondria and triggering of RNA sensors in the cytosol which could inhibit viral protein synthesis. However, mutating the GRSF1 binding site in viral mRNA reduced translation efficiency and cytosolic accumulation of viral mRNA as well, highlighting the role of GRSF1 in IAV mRNA translation.

The human *GRSF1* gene encodes for an RNA-binding protein within the hnRNP family and there exists at least two major isoforms[57]. Both known isoforms of GRSF1 share three quasi-RNA recognition motifs (qRRM) and an acid-rich domain, which is considered to play a regulatory role[58]. The three qRRMs in

GRSF1 mediate interactions with guanine rich sequences of RNA, including cellular mRNAs for glutathione peroxidase 4 (GPx4) and unusual SNARE of the endoplasmic reticulum (Use1)[33,34]. GRSF1 binding to GPx4 and Use1 mRNAs positively regulated their translation, although the exact mechanism is not yet clear. However, the guanine rich sequences of RNA which are bound by GRSF1 have the propensity to form G-quadruplex structures which are proposed to regulate RNAs through a variety of mechanisms[59]. Recently, it has been shown that GRSF1 binding to G-quadruplex structures results in the melting of these RNA secondary structures[56]. This presents one potential mechanism of GRSF1 enhancement of translation, melting of secondary structure in RNA to increase the rate of translation initiation or ribosome processivity. In support of its role in regulating translation, GRSF1 overexpression in mouse embryonic fibroblasts has been shown to increase the amount of GPx4 mRNA found associated with polysomes[34]. Additionally, GRSF1 has been shown to interact with miRNA-346, which regulates the

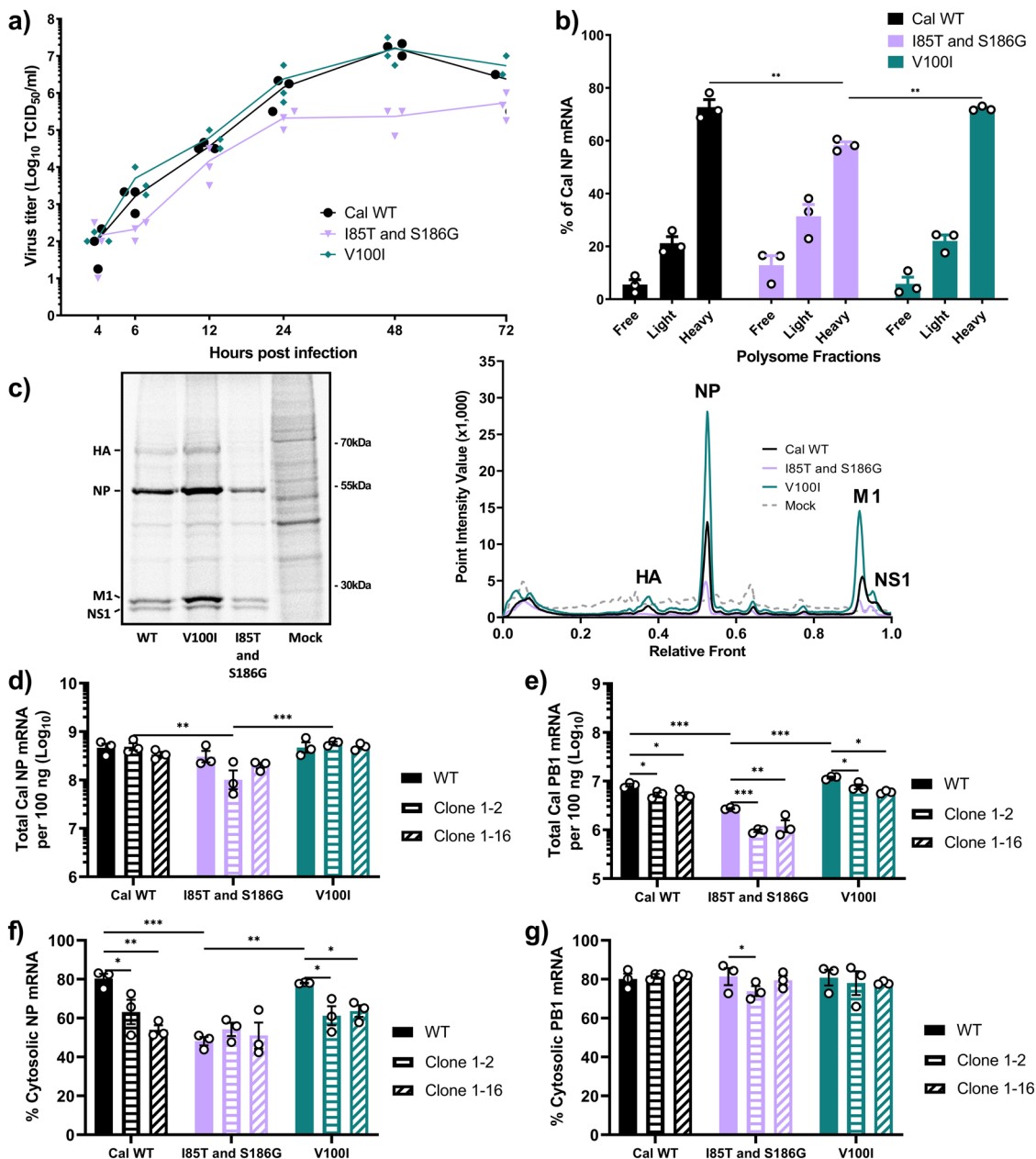

**Fig. 8 GRSF1 and adaptive mutations in PA regulate viral growth, cytosolic mRNA levels, and protein expression. a** Calu-3 cells were infected with the indicated viruses at an MOI of 0.01. At indicated times, virus titers in the supernatant were determined. **b** 293 T cells were infected with the indicated viruses at an MOI of 3 for 1 h. At 4 hpi, cells were lysed and polysome fractionation was performed. NP mRNA in each fraction was quantified by qRT-PCR and fractions are grouped together as in Fig. 1. **c** A549 cells were infected with the indicated viruses at MOI of 3. At 4.5 hpi, cells were labeled with [$^{35}$S] Met/Cys for 0.5 h and lysates were resolved by SDS-PAGE. Gel is a representative image from 3 biological replicates. Densitometric traces of labeled proteins from 3 biological replicates were quantified by Quantity One software. **d**–**g** 293 T WT or 293 T shRNA GRSF1 clone 1–2 or 1–16 were infected with the indicated viruses at an MOI of 5. At 4.5 hpi, cells were lysed and fractionated into nuclear and cytosolic fractions. **d**, **e** Total NP or PB1 mRNA as determined by qRT-PCR and standard a curve. **f**, **g** Percent cytosolic NP and PB1 mRNAs as determined by qRT-PCR. Error bars show means plus/minus standard deviations ($n = 3$ biological replicates). Two-way ANOVA followed by Tukey's multiple comparison test *$P < 0.05$, **$P < 0.01$, ***$P < 0.001$.

translation of hTERT mRNA. Overexpression of miRNA-346 leads to enhanced recruitment of hTERT mRNA to ribosomes, but knockdown of GRSF1 expression abrogates this effect, which suggests a role for GRSF1 in ribosome recruitment via miRNA-346[60].

Previous studies conducted over two decades ago showed that GRSF1 binds to the $^{10}$AGGGU$^{14}$ sequence in the 5′ UTR of the IAV NP and NS mRNAs, and enhances their translation as determined by in vitro assays[31,61]. Our data here indicated that binding of GRSF1 to IAV mRNAs also increased cytosolic levels of the transcripts. In support of GRSF1 possibly being able to regulate the nuclear export of viral mRNAs, a recent Bio-ID screen identified GRSF1 as an interacting partner of the nuclear export factor Aly/REF[62]. This is in line with the known functions of other hnRNPs in nucleocytoplasmic shuttling[63]. Our data here clearly show that loss of GRSF1 binding or knockdown of GRSF1 reduces cytosolic levels of viral mRNA and subsequently leads to a decrease in translation, having major consequences for viral growth and replication. All together, these data suggest that IAV utilizes GRSF1 for selective and rapid expression of viral proteins.

There are major differences in the sequences of human and chicken (*Gallus gallus*) GRSF1 (Supplementary Fig. 5). The three qRRM domains show a high degree of conservation (67%, 60%, and 60% identity) compared to only 43% identity overall for the entire protein. Importantly, the acid-rich regulatory domain only shares 21% identity. This domain is considered to be responsible for functional regulation through interactions with partner proteins[64,65]. Considering our findings here show that PA host-adaptive mutations T85I and G186S affect cytosolic mRNA levels and translation efficiency, it is reasonable to speculate that PA is involved in recruitment and/or attachment of GRSF1 to viral transcripts in a species-specific manner. Residues 85 and 186 are located within the endonuclease domain of PA, closely positioned to where cap-snatching and transcription of the 5′ UTR of viral mRNAs is initiated (Supplementary Fig. 6). While IAV cap-snatching does introduce an additional unique 10–13 nucleotides to the 5′ end of each viral mRNA, the high variability observed in these cap-snatched sequences makes it difficult to predict how cap-snatching could affect GRSF1 binding to viral mRNAs[66,67]. Additionally, NS mRNA is spliced into both NS1 and NS2 and both mRNAs share the 5′ UTR which includes the GRSF1 binding sequence. Therefore, it is possible that GRSF1 regulates the cytosolic levels and translation of NS2 mRNA as well.

The GRSF1 binding sequence that we investigate here, $^{10}$AGGGU$^{14}$, is highly conserved in the 5′ UTRs of mRNAs from the NP and NS genes from both avian and human IAVs, suggesting that both human and avian IAVs utilize GRSF1 (Supplementary Figs 2 and 3). The 5′ UTR of HA mRNA from avian and human IAVs contains the sequence $^{10}$AGGGG$^{14}$which is the same consensus binding sequence that GRSF1 was shown to bind in GPx4 mRNA[34] (Supplementary Figs 2 and 3). This also implicates GRSF1 in the regulation of HA mRNA translation. Importantly, the IAV HA, NP and NS1 proteins are the most rapidly and highly expressed proteins during viral infection[68]. GRSF1 binding to these transcripts provides a potential mechanistic explanation for this. IAV is known to regulate the kinetics and accumulation of viral mRNAs and proteins during the course of infection[69]. How exactly IAV accomplishes this regulation for all eight different vRNA segments is unclear. However, our data suggest that enhanced translation of NP and NS mRNAs conferred by GRSF1 accelerates virus replication and rapid growth in host cells. NP and NS1 are two highly abundant viral proteins in infected cells and play key roles during early stages of viral infection. Compared to components of the vRdRp, large amounts of NP are required for encapsulating positive and negative sense cRNAs and vRNAs to facilitate replication[70]. Rapid expression of NS1 is also required during viral infection to effectively impair host anti-viral responses, accelerate viral replication, and selectively enhance the translation of other viral mRNAs[71,72]. These facts stress the importance of GRSF1 regulation of NP and NS expression to allow for efficient viral replication.

Our data reported here strongly suggest that host-adaptive mutations in PA enable GRSF1 to enhance the cytosolic accumulation and translation of a subset of viral mRNAs. This GRSF1 mediated selective expression of viral proteins represents an additional approach for adaptation of IAVs to mammalian hosts.

## Methods

**Viruses and cell culture**. MDCK (ATCC: CCL-34), 293 T (ATCC: CRL-3216), A549 (ATCC: CRM-CCL-185) and Calu-3 (ATCC: HTB-55) cells were maintained in Dulbecco's modified Eagle's medium (DMEM, Gibco) supplemented with 8% FB Essence (Seradigm) and 50 μg/ml gentamicin (Gibco). A/Michigan/272/2017 (H1N1) was acquired from the International Reagent Resource (FR-1615) and propagated in MDCK cells. Recombinant A/California/04/2009 (H1N1) with mutations I85T and S186G or V100I were rescued in 293 T/MDCK cell co-culture and propagated in eggs[23,73]. Rescued viruses were sequenced for confirmation.

Viral infections were carried out in DMEM containing 0.15% bovine serum albumin (DMEM-BSA).

**Plasmids**. Nan and Cal PA, PB1, PB2 and NP genes were previously cloned into pCAGGS and pPolI expression vectors by use of specific restriction enzymes[20]. Mich PA, PB1, PB2 and NP were cloned into pPolI and pCAGGS similarly using RNA from infected cell lysates. The pPolI-NP-Luc construct, which contains firefly luciferase under the control of the human RNA polymerase I promoter, was obtained from T. Wolff (Robert-Koch Institute, Berlin, Germany). Overlapping site-directed mutagenesis was used to mutate the 5′ UTR sequence from $^{10}$AGGGU$^{14}$ to $^{10}$AGGCA$^{14}$. Chimeric Nan/Cal PA, and Nan PA T85I, G186S, T85I/G186S mutant cDNAs in pCAGGS vectors were created previously[23,36]. Cal PA cDNAs with mutations in I85T, V100I, S186G, and I85T/S186G were created by overlapping site-directed mutagenesis. pCAGGS Nan PA Frameshift (FS) was created by overlapping site-directed mutagenesis to remove the frameshifting site by introducing silent mutations. pCAGGS constructs with mutations in Nan PA and Nan PA FS were created by overlapping site-directed mutagenesis or subcloning. pCAGGS Cal PA-X (without tag) was created by cloning the PCR product produced by specific primers from pCAGGS Cal PA-X-flag as a template[48]. pCAGGS Nan PA-X was created by overlapping site-directed mutagenesis to remove cytosine 598 and then amplifying the coding region for Nan PA-X and cloning into empty pCAGGS[48]. Mutations were introduced in pCAGGS Cal PA-X and pCAGGS Nan PA-X by subcloning. All plasmids were confirmed by sequencing.

**Transfection-based reporter gene assays**. Reporter gene assays were conducted in 293 T cells in 12-well plates. Cells were transfected with 0.4 μg of pCAGGS plasmids containing PA, PB1, PB2 and NP along with 0.1 μg pPolI-NP-luc or pPolI-NP-luc-CA using Lipofectamine 2000 (Invitrogen) in Opti-MEM (Gibco) for 24 h at 37 °C. Luciferase activity was measured by the dual-luciferase reporter assay system (Promega) according to manufacturer protocol. For measuring total mRNA, cell lysates were treated with TRIzol Reagent (Invitrogen) and RNA was extracted according to the manufacturer's protocol. The RNA concentration and purity was determined with a Nanodrop 2000 (Thermo fisher). For RT-PCR reaction, 1 μg of total RNA was digested with DNase I (New England Biolabs) and then purified with TRIzol reagent. 100 ng of purified RNA was used for RT-PCR using RevertAid Reverse Transcription Kit (Thermo Fisher). Real-time PCR was carried out using SYBR Green PCR Master Mix (Applied Biosystems) with an Applied Biosystems 7300 Real-time PCR system. DNA standards for NP-Luc mRNA, Cal NP mRNA, Cal PB1 mRNA, GAPDH mRNA, and U6 snRNA were generated by PCR. A standard curve from $10^9$ to $10^1$ copies was used for quantification. Primers used for RT-PCR and qPCR were designed to be strand specific for mRNA and validated as described by Kawakami et al.[74]. For measuring PA-X shutoff activity, 293 T cells in 12-well plates were transfected with 0.4 μg of pCAGGS-FF-Luc and 0.4 μg of pCAGGS PA-X plasmids or empty pCAGGS vector lipofectamine 2000. At 24 h post transfection, cells were lysed and luciferase activity was measured by the dual-luciferase reporter assay system.

**Western blotting and reagents**. For immunoblotting, cell lysates were mixed with 4X NuPAGE LDS sample buffer and 5% beta-mercaptoethanol. Lysates were resolved by SDS-PAGE and transferred to 0.45 μM PVDF membranes. For detection of protein, primary antibodies used were mouse anti-firefly luciferase (1:1000 dilution, MA1-12556, Invitrogen), mouse-anti PA (1:500 dilution, F5-32[75]), mouse anti-PB1 (1:500 dilution, F5-10[75]), mouse anti-beta actin (1:10,000 dilution, 8H10D10, Cell Signaling Technology), rabbit anti-Lamin A/C (1:1000 dilution, 2032, Cell Signaling Technology), mouse anti α-Tubulin (1:500 dilution, 2144, Cell Signaling Technology), and rabbit anti-GRSF1 (1:3000 dilution, A305-136A, Bethyl Laboratories). Secondary antibodies used were rabbit anti-mouse IgG HRP-linked (1:5000 dilution, 7076, Cell Signaling Technology), goat anti-rabbit IgG HRP-linked (1:5000 dilution, 7074 Cell Signaling Technology) and goat anti-mouse IgG1 HRP-linked (1:3000 dilution, ab97240, abcam).

**Transfection based polyribosome fractionation**. 293 T cells in 100 mm dishes were transfected with 5.3 μg each of pCAGGS plasmids containing PA, PB1, PB2 and NP along with 1.3 μg pPolI-NP-Luc and Lipofectamine 2000 in Opti-MEM for 24 h at 37 °C. Polysome fractionation was carried out as described by Panda et al.[39]. Briefly, at 24 h post transfection, cycloheximide (CHX, Thermo Fisher) was added to cells at a final concentration of 100 μg/mL and incubated for 10 min. Cells were washed three times with ice cold PBS containing 100 μg/mL CHX. Next, 600 μL of high salt lysis buffer (300 mM NaCl, 20 mM Tris-HCl pH 7.5, 10 mM MgCl$_2$, 1x HALT protease inhibitor, 120 units RNAseOUT, and 100 μg/mL CHX) was applied directly to the plate and cell lysates were transferred to microcentrifuge tubes. After incubation on ice for 15 min with occasional mixing and vortexing, the lysates were centrifuged at 12,000 × g for 10 min. Supernatants were transferred to fresh tubes, and total RNA concentration was determined by a NanoDrop 2000. A total of 300 μg of RNA was loaded on top of a 10–50% linear sucrose gradient prepared in high salt lysis buffer and centrifuged for 90 min at 39,000 rpm at 4 °C using a Beckman Coulter SW 41Ti rotor. Gradients were fractionated using a BR-188 Density Gradient Fractionation System (Brandel) with absorbance measured at

254 nm. Gradients were separated into 12 fractions. RNA in each fraction was extracted by TRIzol and precipitated overnight with isopropanol at −20 °C. RT-PCR was carried out as described above. Analysis of mRNA distribution amongst polysome fractions was determined as described by Panda et al.[39].

The equation used for determining the relative amount of RNA per fraction is as follows

$$100*((2^{(cTTube1-cTTubeX)})/(sum: cT\,Tube\,1 - cT\,Tube\,X_1\,through\,X_N)). \quad (1)$$

**Nuclear/cytosolic fractionation.** At the indicated times cells were briefly treated with trypsin for detachment and were collected by centrifugation at 600 x g for 5 min at 4 °C. Fractionation was performed using the Nuclear/Cytosol Fractionation Kit (K266, BioVision) according to the manufacturer's protocol. After fractionation, RNA was isolated by TRIzol Reagent as described above or cell lysates were processed for immunoblotting as described above. RNA was used for qRT-PCR as described above.

$$Percent\,cytosolic\,mRNA\,was\,determined\,as\,\%\,cytosolic$$
$$= (cytosolic\,mRNA/(nuclear\,mRNA + cytosolic\,mRNA))*100. \quad (2)$$

**shRNA GRSF1 knockdown cell lines.** To create a stable GRSF1 knockdown cell line, 293 T cells were transfected with pGIPZ lentiviral shRNA vector containing the shRNA targeting sequence ATGTCAACTATATTCAGTC (RHS4430-200225085 Clone Id: V3LHS_634501(ORF)) obtained from Horizon Discovery. After transfection, cells were cultured in the presence of normal cell growth media for 24 h prior to supplementation with increasing concentrations of puromycin. Single cell colonies were selected based on GFP expression and puromycin resistance and further expanded. GRSF1 protein levels were determined by immunoblotting using rabbit polyclonal anti-GRSF1 (1:3000 dilution, A305-136A, Bethyl Laboratories).

**RNA Immunoprecipitation.** 293 T cells were transfected in 100 mM dishes as described above for transfection based polysome fractionation assays. At 24 h post transfection, cells were lysed with RIPA buffer (25 mM Tris-HCl pH 8.0, 150 mM NaCl, 1% NP-40, 1% Triton X-100, 0.5% Sodium Deoxycholate, 0.1% SDS, and EDTA-free protease inhibitor tablets (A32955, Pierce)). Cell lysates were centrifuged at 10,000 x g at 4 °C, and the supernatant was collected. Five percent of input lysate was reserved for RNA extraction and qRT-PCR to calculate the input amounts. Then, 2 µg rabbit polyclonal anti-GRSF1 (A305-136A, Bethyl Laboratories) or 2 µg rabbit IgG isotype control (02-6102, Invitrogen) was added to 1 mg lysate and incubated at 4 °C with rotation for 2 h. After 2 h, 0.5 mg Dynabeads Protein G (10003D, Invitrogen) were added to each sample and further incubated at 4 °C for 1 h with rotation. Protein/RNA complexes were eluted by magnetic separation and re-suspending in either 4X NuPAGE LDS sample buffer and 5% beta-mercaptoethanol or TRIzol Reagent for RNA extraction. qRT-PCR was carried out as above for detection of mRNAs.

**Viral infections.** For polyribosome fractionation, 293 T cells were infected with the indicated viruses at an MOI of 3 for 1 h at 37 °C in DMEM containing 0.15% bovine serum albumin (DMEM-BSA). At the indicated times after infection, 100 µg/mL CHX was added to cells and cultured for 10 min. Cell lysate preparation, ultracentrifugation, and fractionation were carried out as above and gradients were separated into 12 fractions. Strand-specific qRT-PCR for Cal NP mRNA was carried out as described above[73,74].

For monitoring protein synthesis, A549 cells were infected with the indicated viruses at an MOI of 3 for 1 h at 37 °C in DMEM-BSA. At the indicated times, cells were metabolically labeled with 50 µCi [³⁵S]Met-Cys for 30 min in medium lacking methionine and cysteine prior to lysis with RIPA buffer. Radiolabeled lysates were resolved by SDS-PAGE. Dried gels were exposed to a phosphor screen and visualized using a Bio-Rad personal molecular imager. To compare the sum of intensities of pixels across lanes, volume analysis was performed using Quantity One 1-D analysis software.

For monitoring virus growth, Calu-3 cells were infected with the viruses for 1 h at 37 °C at an MOI of 0.01 and cultured with DMEM-BSA supplemented with acetylated trypsin at 2 ug/mL. At the indicated time points, cell supernatants were collected and titrated in MDCK cells[76].

**Sequence analysis.** Influenza A virus sequences for A/California/04/2009(H1N1), A/Michigan/272/2017(H1N1), and A/Chicken/Nanchang/3-120/01(H3N2) were from fludb.org and correspond to NCBI:txid641501, NCBI:txid2033552 and NCBI:txid215853. For comparison of residue V100I, protein sequences for PA from 2009 pH1N1-like influenza A viruses (Human Host) were analyzed from publically available sequencing data on fludb.org using the Analyze Sequence Variation (SNP) program. For comparison of protein sequences for *Homo sapiens* GRSF1 (NCBI Reference Sequence: NP_002083.4) and *Gallus gallus* GRSF1 (NCBI Reference Sequence XP_015131904.2), the program Clone Manager (Sci Ed Software) was used. Comparison of the 5' UTR sequences of IAV mRNAs was performed by downloading complete (coding and non-coding) sequences for all 8

gene segments from fludb.org for both human and avian hosts. Sequences were then trimmed to the length of the 5' UTR for each segment, PB2 = 27 nucleotides, PB1 = 23 nucleotides, PA = 23 nucleotides, HA = 32 nucleotides, NP = 45 nucleotides, NA = 19 nucleotides, M = 25 nucleotides, NS = 26 nucleotides, and aligned for LOGO analysis using a custom R script with Biostrings and ggseqlogo.

**Statistics and reproducibility.** A minimum of three biological replicates was performed for each experiment. Error bars shown represent the means plus/minus the standard deviations. Statistical analysis was performed using Graph Pad Prism 9.0 software. Comparisons between multiple groups were performed using a 1-way or 2-way ANOVA with Tukey's post-test. Comparisons between single groups was performed using a student's *t*-test.

**Reporting summary.** Further information on research design is available in the Nature Research Reporting Summary linked to this article.

## Data availability
All raw and processed data are available at request from the corresponding author. Source data for main and supplementary figures are provided as Supplementary Data 1. All uncropped and unedited blot/gel images are available in Supplementary Fig. 7.

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

## Acknowledgements

A/Michigan/272/2017(H1N1) was provided by The International Reagent Resource. This research was funded by National Institute of Health through Grant number R01AI129988. M.L. and J.S. were supported by National Institutes of Health Grant number T32 GM135134. The contents of this study do not necessarily represent the official views of the NIH. We thank members of the Takimoto lab for critical reading of the manuscript. We also thank Katrin Dahm for excellent technical expertize.

## Author contributions

M.L., J.S., and T.T. conducted the experiments. M.L. and T.T. designed the experiments. T.T. supervised the work. M.L. and T.T. wrote the original draft.

## Competing interests

The authors declare no competing interests.
