## [Peer Review File · Communications Biology]

Reviewers' comments:

Reviewer #1 (Remarks to the Author):

Birds are the natural reservoir for influenza viruses. Adaptive mutations are required to allow these viruses to replicate and spread in humans. These host adaptive mutations have been extensively studied, but precise mechanisms that confer advantage in humans remain only partially understood. Here, Lutz, et al., used the IAV replicon system to study how adaptive mutations in polymerase subunits affect viral mRNA synthesis, nuclear export and translation. They report that adaptations in the PA subunit from 2009 pH1N1 enhanced nuclear export and translation of viral mRNA, which was dependent on the host mRNA binding protein GRSF1.

This manuscript is well written and the data is clearly presented. The data in Figures 1 to 4 that thoroughly mapped PA residues involved in regulating viral mRNA export and translation is quite convincing. However, the manuscript is limited in scope. The precise role of the host mRNA transport factor GRSF1 was incompletely characterized both in the replicon system and inauthentic IAV infection and the data does not properly support a role for GRSF1 in this system. This aspect of the manuscript needs to be strengthened to fully support the authors' conclusions.

Specific comments:

The Introduction provides relevant background information, and yet, adaptive mutations that affect interactions between IAV polymerase subunits and host factors were only superficially described. The reader would benefit from some of the essential and fascinating examples of host restriction and IAV adaptive mutation, which would provide good context for this study and the GRSF1 finding.

The data presented in Figures 1-3 clearly and methodically takes the reader through experiments that lead to the conclusion that PA substitutions T85I, V100I and G186S increase the production of luciferase from the pPOL1-NP-luc replicon plasmid, with V100I causing the greatest increase in luciferase production per mRNA. The use of a Table charting the change in the identity of residue 100 in circulating H1N1 strains from 2009 to 2020 was particularly effective. Ruling out PA-X activity in the mechanism of action of the V100I construct is very important and should be included in the main dataset rather than the supplement, because PA-X will immediately come to mind for any reader with familiarity with IAV host shutoff.

Figure 4: While the fractionation experiments make it clear that the PA adaptive mutations affect accumulation of the luciferase mRNA in the cytoplasm, the V100I substitution did not have a significant impact on its own in these assays. This stands apart from the data in Figure 3 that shows that V100I had the strongest positive effect on luciferase production. The authors should be more cautious in their interpretation of the fractionation experiments in Figure 4 and acknowledge the lack of a significant change for V100I, and provide some interpretation for this finding. Furthermore, while the inclusion of GAPDH mRNA as a negative control is good, this data could be further strengthened by probing for a snRNA that remains restricted to the nucleus.

Figure 5: Here, in analyzing the potential effects of GRSF1 binding on the nucleocytoplasmic transport of the luciferase reporter mRNA, the authors claim that a 2 nucleotide difference in the GRSF1 binding site reduced the translation efficiency of vRdRp transcribed mRNAs. This conclusion is not properly supported by the data. No experiment was performed to confirm that the 2 nucleotide change actually reduced GRSF1 binding, which undermines the experiments that follow. However, most importantly, the data shows that the 2 nucleotide change only moderately reduces the accumulation of the luciferase enzyme in the system (Fig. 5E). It is confusing that the authors continue to focus on the replicon system with mismatched Cal PA subunit here after having already narrowed the mechanism of nuclear export regulation to a few amino acid substitutions (Fig. 4). This takes us further away from mechanistic understanding.

Figure 6: Here, the authors analyze the effect of GRSF1 silencing on nuclear export and translation of luciferase mRNAs. They show that in the context of the mismatched Pol subunits with Cal PA, GRSF1 silencing negatively impacts luc mRNA transport and luc activity. However, the effects of GRSF1 silencing on V100I or the other key amino acid substitutions in PA were not investigated, and more importantly, the effect of GRSF1 silencing on IAV replication was not investigated. This experiment is achievable and would greatly increase our understanding of GRSF1's role in viral mRNA export in authentic infection.

Figure 7: These data show that the PA substitutions only have a minor effect on the accumulation of NP mRNAs in the cytoplasm, which again, does not strongly support a role for GRSF1 in IAV replication. Considering that only NP and NS have previously characterized GRSF1 binding sites, it would be useful to compare multiple viral mRNAs that possess or lack GRSF1 binding sites in this system.

Additional questions: The authors indicate that the GRSF1 binding motif AGGU is found in the 5' end of NP and NS1 mRNAs. Is this motif found in other IAV mRNAs? How conserved is it? Given that there are very few IAV mRNAs, it would be straightforward and informative to survey all of them. Also, does variability at the 5' end of viral mRNAs due to cap-snatching affect GRSF1 binding? How does viral pre-mRNA splicing affect the outcome? Addressing these concepts through experimentation and discussion would strengthen the manuscript.

Reviewer #2 (Remarks to the Author):

The manuscript (COMMSBIO-22-1683-T) entitled "Host adaptive mutations in the 2009 H1N1 pandemic influenza A virus PA gene regulate translation efficiency of viral mRNAs" by Lutz et al is an interesting and well organized research article. The authors could successfully unmask the molecular mechanism(s) by which adaptive PA protein from 2009 pH1N1 (Cal) virus could enhance its viral polymerase activity. The authors could show that the 2009 pH1N1 PA, and the associated host adaptive mutations, led to greater translation efficiency. This was due to enhanced nuclear export of viral mRNA, which was dependent on the host RNA binding protein GRSF1. Figures and Tables are clear and well presented. References are adequate and updated. I consider this manuscript of interest, however, I have some comments:

1. Despite that the PA, PB2 and PB1 are in complex, forming the viral ribonucleoprotein complex (vRNP) and that the three viral polymerases are interacting with each other. Moreover, the authors did prove that the N-terminal part in the PA is responsible for the observed viral polymerase activity and related mechanisms to do so. Despite that the N-terminal of PA is well known to interact directly with PB1 subunit, the authors ignored the role of 2009 pH1N1 PB1 in this study. In previous studies (e.g. DOI 10.1099/jgv.0.000390), the 2009 pH1N1 PB1 was found to even improve the polymerase activity of well-adapted avian and mammalian strains.

2. Line 50: is it mRNA splicing or spicing?

Reviewer #3 (Remarks to the Author):

In this manuscript, xxx et al. examine the differences between an avian and a human influenza A virus polymerase in human cells to understand how changes to the polymerase can improve human adaptation. They find that the source of the PA subunit of the polymerase changes the translation efficiency of the synthesized mRNA, due to altered recruitment of the GRSF1 RNA binding protein, which in turn changes mRNA export. These changes can affect viral replication. This is an interesting study presenting a novel aspect of the species specificity of viral polymerases that will need to be

considered in future studies and characterized better. The data are compelling and well analyzed, and the study is well done. I have a few minor comments, including one involving a new experiment:

- 1) A key part of the study is the translation efficiency measurements, the luciferase activity to mRNA molecules ratio. However, there are no error bars on any of these measurements, suggesting they are ratio of the average values. It would be better if the authors averaged the values per experiment and then plotted the average of the ratios. This would allow the authors to do statistics and the reader to assess the variation per experiment. (also on line 256 the authors state that the translation efficiency was "significantly reduced" but they have no statistics on the plot).
- 2) While they see phenotypes with the point mutations described in Fig. 2, the effect is much less pronounced than the full Nan PA. In light of this, I think the authors should moderate a bit the language of their conclusions (lines 181-182) by adding the word "contribute to" in the sentence. Otherwise, their conclusion implies the 3 residues explain the entire effect.
- 3) To fully exclude the effect of PA-X, the authors should really compare the Nan FS to the Cal FS. This is especially true since the mutants they test show much smaller differences in translation efficiency than the full proteins.
- 4) As a control, it would be good to show that GRSF1 no longer binds to the CA mutant construct.
- 5) Line 289 - I would remove the "greatly" here since the cytoplasmic RNA goes from 60% to 45%.
- 6) Why do the authors use custom values for the P value asterisks instead of the conventional ones? (* = $P \leq 0.05$, ** = $P \leq 0.01$, *** = $P \leq 0.001$). The conventional ones would be better for readability.

Responses to the reviewers' comments:

We thank all the reviewers for their critical reading of the manuscript and thoughtful suggestions. Accordingly, we have addressed the comments of the reviewers and modified the manuscript with new data and clarification in key areas. We hope our modified manuscript meets the expectations of both the reviewers and editors and have outlined below the specific changes.

Major points:

Reviewer 1:

- “The Introduction provides relevant background information, and yet, adaptive mutations that affect interactions between IAV polymerase subunits and host factors were only superficially described. The reader would benefit from some of the essential and fascinating examples of host restriction and IAV adaptive mutation, which would provide good context for this study and the GRSF1 finding.”
 - Accordingly, we included additional background information regarding known host restriction factors and adaptive mutations in the vRdRp in Introduction lines 36-49.
- “Ruling out PA-X activity in the mechanism of action of the V100I construct is very important and should be included in the main dataset rather than the supplement, because PA-X will immediately come to mind for any reader with familiarity with IAV host shutoff.”
 - We agree and have included this supplemental figure in the main text of the manuscript as Figure 4 now.
- “Figure 4: While the fractionation experiments make it clear that the PA adaptive mutations affect accumulation of the luciferase mRNA in the cytoplasm, the V100I substitution did not have a significant impact on its own in these assays. This stands apart from the data in Figure 3 that shows that V100I had the strongest positive effect on luciferase production. The authors should be more cautious in their interpretation of the fractionation experiments in Figure 4 and acknowledge the lack of a significant change for V100I, and provide some interpretation for this finding. Furthermore, while the inclusion of GAPDH mRNA as a negative control is good, this data could be further strengthened by probing for a snRNA that remains restricted to the nucleus.”
 - The reviewer makes an excellent point. We have provided additional experimental data in the revised (now) Figure 5. Fig 5d shows that in the context of the Nan vRdRp, the V100I mutation does provide a slight increase to the accumulation of luciferase mRNA in the cytoplasm. However, in conjunction with our data in Figs 6, 7, and 8, we agree that V100I may not strongly contribute to cytosolic levels of viral mRNAs, although it enhances translation efficiency and may still play a role in host adaptation. We have updated the language throughout the manuscript to focus the conclusions on mutations T85I and G186S. We have also provided new experimental evidence in Figure 5b showing the cytosolic levels of U6 snRNA.
- “Figure 5: Here, in analyzing the potential effects of GRSF1 binding on the nucleocytoplasmic transport of the luciferase reporter mRNA, the authors claim that a 2 nucleotide difference in

the GRSF1 binding site reduced the translation efficiency of vRdRp transcribed mRNAs. This conclusion is not properly supported by the data. No experiment was performed to confirm that the 2 nucleotide change actually reduced GRSF1 binding, which undermines the experiments that follow. However, most importantly, the data shows that the 2 nucleotide change only moderately reduces the accumulation of the luciferase enzyme in the system (Fig. 5E). It is confusing that the authors continue to focus on the replicon system with mismatched Cal PA subunit here after having already narrowed the mechanism of nuclear export regulation to a few amino acid substitutions (Fig. 4). This takes us further away from mechanistic understanding.”

- To illustrate the difference in GRSF1 binding conferred by the CA mutation in pPoll-NP-luc, we have repeated the RNA immunoprecipitation experiments using both the CA mutant and original template. This new data showing the specific binding of GRSF1 to the AGGGU template is described in lines 236-251 and shown in Figure 6b. Also, to avoid confusion by continuing to use the mismatched Cal PA subunit, we have performed additional experiments using the Cal vRdRp with Nan PA or Nan PA mutants using the CA mutant template. The new data support our conclusion that mutations T85I and G186S require GRSF1 binding to viral mRNAs for enhanced translation efficiency and cytosolic accumulation. These data are described in lines 251-266 and shown in Figure 6c-f. The previous data is now shown in supplementary Figure 4.
- “Figure 6: Here, the authors analyze the effect of GRSF1 silencing on nuclear export and translation of luciferase mRNAs. They show that in the context of the mismatched Pol subunits with Cal PA, GRSF1 silencing negatively impacts luc mRNA transport and luc activity. However, the effects of GRSF1 silencing on V100I or the other key amino acid substitutions in PA were not investigated, and more importantly, the effect of GRSF1 silencing on IAV replication was not investigated. This experiment is achievable and would greatly increase our understanding of GRSF1’s role in viral mRNA export in authentic infection.”
 - We have performed additional experiments using the Cal vRdRp with Nan PA or Nan PA mutants in the GRSF1 knockdown cells. These new data support our conclusion that mutations T85I and G186S require GRSF1 for enhanced translation efficiency and cytosolic accumulation of viral mRNA. The new data are described in lines 267-284 and shown in the new Figure 7. Also, we infected GRSF1 knockdown cells with the Cal WT, I85T and S186G, and V100I viruses and compared NP and PB1 mRNA levels. The results indicate GRSF1 KD specifically reduced cytoplasmic accumulation of NP mRNA. These new data are described in lines 304-318 and shown in new Figure 8d-g.
- “Figure 7: These data show that the PA substitutions only have a minor effect on the accumulation of NP mRNAs in the cytoplasm, which again, does not strongly support a role for GRSF1 in IAV replication. Considering that only NP and NS have previously characterized GRSF1 binding sites, it would be useful to compare multiple viral mRNAs that possess or lack GRSF1 binding sites in this system. “
 - To further show the role of GRSF1 during viral infection we performed additional experiments in GRSF1 knockdown cells as described above. In addition to characterizing NP mRNA, we also examined PB1 mRNAs, which are predicted to not utilize GRSF1 as

they share the sequence with our CA mutant pPoll NP Luc construct. The results showed that cytosolic levels of PB1 mRNA was not affected by GRSF1 knockdown or I85T and S186G mutations.

- “Additional questions: The authors indicate that the GRSF1 binding motif AGGU is found in the 5’ end of NP and NS1 mRNAs. Is this motif found in other IAV mRNAs? How conserved is it? Given that there are very few IAV mRNAs, it would be straightforward and informative to survey all of them. Also, does variability at the 5’ end of viral mRNAs due to cap-snatching affect GRSF1 binding? How does viral pre-mRNA splicing affect the outcome? Addressing these concepts through experimentation and discussion would strengthen the manuscript. “
 - o We performed sequence analysis of the 5’ UTR of all IAV mRNAs from human and avian IAVs now included as supplementary Figures 2 and 3. The sequence AGGGU in NP and NS mRNAs is highly conserved amongst both avian and human IAVs. Also, HA mRNAs from avian and human IAVs have a conserved AGGGG sequence which could also regulate GRSF1 binding. We have discussed this in the revised manuscript in lines 374-381. Considering the high variability observed in cap-snatched sequences by Koppstein et. Al (doi: 10.1093/nar/gkv333) and Gu. et. Al (doi: 10.1261/rna.054221.115), it is hard to predict how cap-snatching would affect GRSF1 binding to viral mRNA transcripts. We address this in lines 366-371. Lastly, it is important to note that NS mRNA is spliced into both NS1 and NS2 and share the same 5’ UTR. We address this in lines 371-373.

Reviewer #2:

- “1. Despite that the PA, PB2 and PB1 are in complex, forming the viral ribonucleoprotein complex (vRNP) and that the three viral polymerases are interacting with each other. Moreover, the authors did prove that the N-terminal part in the PA is responsible for the observed viral polymerase activity and related mechanisms to do so. Despite that the N-terminal of PA is well known to interact directly with PB1 subunit, the authors ignored the role of 2009 pH1N1 PB1 in this study. In previous studies (e.g. DOI 10.1099/jgv.0.000390), the 2009 pH1N1 PB1 was found to even improve the polymerase activity of well-adapted avian and mammalian strains. “
 - o We acknowledge that the data we have presented in the paper here does not characterize the role of pH1N1 PB1 in vRdRp activity. However, previous work from our lab (doi: 10.1128/JVI.00522-11) has shown that pH1N1 PB1 from Cal does not significantly alter overall vRdRp activity and unpublished work in our lab has seen that PB1 from Cal or Mich does not significantly affect vRdRp transcription of mRNA in the Nan vRdRp backbone.
- “2. Line 50: is it mRNA splicing or spicing? “
 - o We thank the reviewer for catching this typo. We have updated the manuscript to say “mRNA splicing”.

Reviewer #3:

- “A key part of the study is the translation efficiency measurements, the luciferase activity to mRNA molecules ratio. However, there are no error bars on any of these measurements,

suggesting they are ratio of the average values. It would be better if the authors averaged the values per experiment and then plotted the average of the ratios. This would allow the authors to do statistics and the reader to assess the variation per experiment. (also on line 256 the authors state that the translation efficiency was “significantly reduced” but they have no statistics on the plot).”

- We thank the reviewer for this suggestion, the original problem and reason for showing the average values of the ratios was that the ratios of the luciferase activity to mRNA copy numbers was not a normally distributed data set (making statistical analysis problematic). To rectify this, we have used statistical software to transform the data into a normalized data set on which statistics were able to be run. We have updated all the figures accordingly.
- “2) While they see phenotypes with the point mutations described in Fig. 2, the effect is much less pronounced than the full Nan PA. In light of this, I think the authors should moderate a bit the language of their conclusions (lines 181-182) by adding the word “contribute to” in the sentence. Otherwise, their conclusion implies the 3 residues explain the entire effect.”
 - We agree with the comments. We have updated the language in the revised manuscript in lines 180-181 to be more cautious with our interpretation.
- “3) To fully exclude the effect of PA-X, the authors should really compare the Nan FS to the Cal FS. This is especially true since the mutants they test show much smaller differences in translation efficiency than the full proteins.”
 - In also addressing reviewer 1’s comments, we have included the PA-X and PA frameshift data in the main article as Figure 4. We included data comparing Cal PA FS with Nan PA FS, which clearly indicate that PA-X is not a factor that affects translation efficiency. We also added data showing the effect of Nan PA mutations in the background of Nan PA FS. These new data are described in lines 195-213.
- “4) As a control, it would be good to show that GRSF1 no longer binds to the CA mutant construct. “
 - In line with the same comment being made by reviewer #1, we have repeated our RNA immunoprecipitation experiments. The results show that GRSF1 binding to the CA mutant is significantly reduced compared to the original template (Figure 6B).
- “5) Line 289 – I would remove the “greatly” here since the cytoplasmic RNA goes from 60% to 45%. “
 - We have provided new data in the manuscript to show the effect of GRSF1 knockdown on the cytosolic levels of NP and PB1 mRNAs in the revised Figure 8. We have updated our language to be more cautious with our interpretations.
- “6) Why do the authors use custom values for the P value asterisks instead of the conventional ones? (* = $P \leq 0.05$, ** = $P \leq 0.01$, *** = $P \leq 0.001$). The conventional ones would be better for readability.”

- We thank the reviewer for this suggestion, originally we were using the automatic p-values provided by GraphPad Prism 9.0 but we have changed the analysis for all figures to utilize the conventional p-values.

REVIEWERS' COMMENTS:

Reviewer #1 (Remarks to the Author):

The authors have done a very thorough job of addressing critiques from all reviewers with text modifications and new experiments.

The authors may consider discussing some of the other fascinating biology that GRSF1 regulates. Specifically, the role that GRSF1 plays in melting G4 structures and participating in the degradation of mitochondrial RNA as part of the mitochondrial degradosome (including hSUV3 and PNPT1).

<https://pubmed.ncbi.nlm.nih.gov/29967381/>

Interfering with GRSF1 expression, as performed in this study in Figure 7, would be predicted to have profound effects on the regulation of mitochondrial RNA levels and release of mitochondrial dsRNA that could activate PKR and hinder viral protein synthesis.

Reviewer #2 (Remarks to the Author):

The authors provided satisfactory answers to all my concerns. I recommend the publication of the manuscript in its current form.

Reviewer #3 (Remarks to the Author):

This manuscript is a resubmission that I previously reviewed. The authors have address all of my concerns/comments from the previous submission and I do not have any further comments.